# A Review of the Popular Uses, Anatomical, Chemical, and Biological Aspects of Kalanchoe (Crassulaceae): A Genus of Plants Known as “Miracle Leaf”

**DOI:** 10.3390/molecules28145574

**Published:** 2023-07-21

**Authors:** Evelyn Assis de Andrade, Isadora Machinski, Ana Carolina Terso Ventura, Sarah Ainslie Barr, Airton Vicente Pereira, Flávio Luís Beltrame, Wendy Karen Strangman, Robert Thomas Williamson

**Affiliations:** 1Pharmaceutical Science Graduate Program, State University of Ponta Grossa, Ponta Grossa 84030-900, PR, Brazil; evelyn.aandrade@gmail.com (E.A.d.A.); isadora.machinski@gmail.com (I.M.); anacarolinatervent@gmail.com (A.C.T.V.); airtonvp@uepg.br (A.V.P.); flaviobeltra@gmail.com (F.L.B.); 2Department of Chemistry and Biochemistry, University of North Carolina Wilmington, Wilmington, NC 28409, USA; sab9514@uncw.edu (S.A.B.); strangmanw@uncw.edu (W.K.S.)

**Keywords:** traditional use, chemical composition, botanical description, pharmacological activities, natural products, bioactive compounds, Kalanchoe, Crassulaceae

## Abstract

Species of the genus Kalanchoe have a long history of therapeutic use in ethnomedicine linked to their remarkable healing properties. Several species have chemical and anatomical similarities, often leading to confusion when they are used in folk medicine. This review aims to provide an overview and discussion of the reported traditional uses, botanical aspects, chemical constituents, and pharmacological potential of the Kalanchoe species. Published scientific materials were collected from the PubMed and SciFinder databases without restriction regarding the year of publication through April 2023. Ethnopharmacological knowledge suggests that these species have been used to treat infections, inflammation, injuries, and other disorders. Typically, all parts of the plant are used for medicinal purposes either as crude extract or juice. Botanical evaluation can clarify species differentiation and can enable correct identification and validation of the scientific data. Flavonoids are the most common classes of secondary metabolites identified from Kalanchoe species and can be correlated with some biological studies (antioxidant, anti-inflammatory, and antimicrobial potential). This review summarizes several topics related to the Kalanchoe genus, supporting future studies regarding other unexplored research areas. The need to conduct further studies to confirm the popular uses and biological activities of bioactive compounds is also highlighted.

## 1. Introduction

The Crassulaceae J. St.-Hil. family is composed of 36 genera [1]. Species of this family are distributed in Africa and Asia, predominantly in Madagascar and Arabia [2,3] but are also found in the Americas and in Australia (Figure 1) [4].

The genus Kalanchoe Adans (Heterotypic Synonyms: Baumgartenia Tratt., Bryophyllum Salisb., Crassuvia Comm. ex Lam., Geaya Costantin and Poiss., Kitchingia Baker, Meristostylus Klotzsch, Physocalycium Vest, and Vereia Andrews) belongs to the Crassulaceae family and comprises 179 accepted species [5]. The synonyms (according to Plants of the World Online, facilitated by the Royal Botanic Gardens) and number of occurrences worldwide (according to Global Biodiversity Information Facility) of the accepted species are shown in Table 1. 

The term *Kalanchoe* was originally used by Michel Adanson in 1763 and it refers to the phonetic transcription of the Chinese term “*Kalan Chauhuy*”, which means “what falls and grows”. The name *Kalanchoe* describes the propagation of leaf embryos. Another explanation for the name relates it to the words “*kalanka*” and “*chaya*”, which are used by Brazilian indigenous people and, respectively, mean stain/rust and shine, alluding to the reddish roots and shiny leaves [3,6]. Figure 2 shows the distribution of the native and introduced species of the genus *Kalanchoe* around the world [7].

Species of this genus are popularly known as “mother-of-thousands” or “mother-of-millions” due to their propagation by leaf embryos [8]. Some members of the genus *Kalanchoe* have a long history of therapeutic use and are known as “miracle leaf” because of their remarkable healing properties and traditional use in the treatment of several diseases and disorders [6,9,10,11,12]. Some of these biological activities have been correlated with specific classes of secondary metabolites already described in the *Kalanchoe* species. Examples include cardioactive glycosides and phenolic compounds (phenolic acids, flavonoids, and tannins) [13].

However, a detailed literature search revealed that only a limited number of species described as “miracle leaf” have anatomical and structural similarities and are used in folk medicine to treat a variety of health problems and disorders. Consequently, this review provides a critical overview of the main aspects published in the literature regarding the traditional uses, botanical characteristics, chemical composition, and pharmacological activity of species of the *Kalanchoe* genus, and aims to contribute to the knowledge of this genus, discussing important biological and chemical aspects described in these studies, and providing material for new evaluation.

## 2. Results and Discussion

### 2.1. Traditional Uses

The genus *Kalanchoe* is widely used in folk medicine to treat different health diseases and disorders. Thus far, only 21 of the 133 species of the genus *Kalanchoe* have been reported regarding their popular uses, as described in Table 2.

From these 21 species, there exists a broad ethnopharmacological knowledge of four species that are more often cited as medicinal plants (*K. pinnata*, *K. laciniata*, *K. crenata*, and *K. daigremontiana*), suggesting that they can be adopted to treat wounds, cancer, diabetes, infections, and inflammation. However, there are no reports in the scientific literature that describe the amounts of plant or dosages for ethnomedicinal uses.

All parts of the *Kalanchoe* species are traditionally used for medicinal purposes, but the juice or crude extract preparations (produced by maceration) are cited as the primary forms of administration [6,8,11,15,18,20,21,28,37,43,49,63,65,70,78,79,81].

In the cases of *K.* × *houghtonii*, *K. flammea*, *K. gastonis-bonnieri*, and *K. integra*, the literature does not describe which parts of the plant, method of preparation, or the dosage are popularly recommended for medicinal use. As is the case with many medicinal plants, folk-information related to traditional use of medicinal plants contributes to the search for scientific basis in these treatment regimens. These data, and the important lack thereof in most cases, reinforce the importance of additional investigations into the chemistry and bioactivity of this genera.

### 2.2. Botanical Description

Species of the Crassulaceae family are herbaceous or sub-shrubs, usually succulent, opposite, or alternate, and exstipulate. The flowers are actinomorphic, hermaphrodite, and usually cymose [5]. Species of the *Kalanchoe* genus are herbaceous or sub-woody; they have small branches and can reach from 1 to 1.5 m in height, especially during their flowering stage. Its leaves are opposite, succulent, oval, and have crenated margins, which are 10 to 20 cm long. Flowers can measure up to 5 cm in length, are pendant, and are arranged in inflorescences. Fruits are membranous, and the seeds are ellipsoid. The stem has thin-walled cells located deep in the epidermis. These cell walls are impregnated with resin, forming a thin layer that can reduce liquid evaporation [94,95,96]. 

These species adapt well and tolerate extreme conditions, such as lighting and water scarcity. One feature of this plant is a compartment in the leaves and stem tissues that can store and inhibit water loss [2,96,97]. This physical adaptation works in tandem with crassulacean acid metabolism (CAM), a metabolic adaptation to perform photosynthetic CO_2_ fixation and water loss reduction. During the night, and at low temperatures, the stomata open, and the plant can assimilate atmospheric CO_2_. However, daylight closes the stomata structure and CO_2_ fixation occurs [98,99,100]. The stomatas have been described in detail and can be considered anatomical markers of the family [101].

Species of the genus *Kalanchoe* are popularly known due to their propagation by leaf embryos, and these propagules (also called leaf bulbs or bulbils) from the margins of the leaves are responsible for their tremendous invasiveness. New plants can be produced from parts of the mother plant, especially by clonal growth through the bulbs that arise from the leaf margins. In suitable open places (such as rocky or sandy environments) these populations can quickly form dense stands. This feature is the primary reason they are popularly known as “mother-of-thousands” or “mother-of-millions” [8,12,102].

Only 16 of the 133 species of the genus *Kalanchoe* have had their botanical aspects formally described in the literature (Table 3). More specifically, 11 of them have a macroscopic description (*K. blossfeldiana*; *K. marmorata*; *K. beharensis*, *K. laxiflora*, *K. orgyalis*, *K. rhombopilosa*, *K. synsepala*, *K. tetraphylla*, *K. tomentosa*, and *K.* × *houghtonii*), and only 5 have additional botanical evaluation/microscopical analysis of the plants (*K. daigremontiana*, *K. delagoensis*, *K. laciniata*, *K. pinnata*, and *K. pumila*). 

In the case of *K. blossfeldiana*, five genotypes were also distinguished by morphological characterization (assessing the flower’s anatomical aspects and plant height), and molecular profiling (random amplified polymorphism DNA (RAPD), inter-simple sequence repeats (ISSR), and start codon targeted (SCoT)-polymerase chain reaction (PCR) tools) [103]. 

**Table 3 molecules-28-05574-t003:** Botanical aspects of *Kalanchoe* species.

Species	Macro Aspects	Micro Aspects	References
*K. beharensis*	The largest species of the genus, with 3 m in height; unbranched stems; leaves crowded at the branch tips; lobed, covered in a dense felt; ranging from 12–35 cm in length and 7–35 cm in width.		[6]
*K. blossfeldiana*	Dark green, succulent, and perennial plant, with scallop-edged leaves and large umbels of flower clusters held above the foliage. The fleshy, dark shiny green leaves have lobed edges and can reach 7.7 cm in length and 3.8 cm in width. Floral colors range from traditional red to yellow, orange, salmon and pink.		[104]
*K. daigremontiana*	Perennial short-lived succulent herb; monocarpic multi-annuals. The most characteristic feature of the species is its method of asexual reproduction by auto-propagation. Flowering tends to be sporadic, in winter, and, when it occurs, the main stalk elongates vertically, developing a terminal inflorescence of small, bell-shaped, pendulous flowers with a pinkish or purple corolla. The stem is unbranched, up to 1.5 m in height. The leaves are thick, fleshy, lanceolate, tapered at the apex and serrated in the margins, dark green colored, and have purple-brown spots on the abaxial side. The apex bears hydathodes and adventitious buds, from which propagules are formed and developed.	The epidermis is single-layered, with parenchymatic cells, convex outer walls surface, wax patches in cuticles, is smooth-undulating, and striated only on subsidiary cells. The leaves are amphistomatic, with anisocytic stomata. The subepidermal mesophyll consists of one or several layers of small, closely adherent cells. The central vascular bundles are surrounded by perivascular sheaths composed of mesophyll cells. Between the epidermis and mesophyll in the petioles there are 1–3 layers of compact angular collenchyma. The vascular bundles are collateral. In the central veins in the petiole and the leaf are three large bundles. The cross-sections show fine lateral vascular bundles surrounding large bundles in the petioles and leaf blades. The different tissues of the leaf contain numerous phenolic idioblasts, accumulating phenolic compounds in their vacuoles, present in epidermal cells, in the subepidermal layer, near the vascular elements, around the large vascular bundles in the leaf petioles, and surrounding the smaller vascular bundles, dispersed in the parenchyma as single cells or form multicellular aggregates.	[6,12,29,97,105,106]
*K. delagoensis*	It has dark purplish, speckled, tubular leaves, which are filled with plantlets. It typically grows to about 1 m in height before blooming. It overwinters as a terminal inflorescence bearing orange or red pendant bell-shaped flowers and then dies.	The leaves are tubular and have 6–8 apical buds. The epidermal cells are uniseriate with sinuous anticlinal walls. The leaves are amphistomatic with anisocytic stomata. The mesophyll has regular chlorenchyma. The vascular system has collateral bundles distributed in the form of an arc. Anthocyanin idioblasts occur throughout the leaf blade, in the epidermis; hypodermis; layer beneath the hypodermis; scattered in the chlorenchyma; surrounding the vascular bundles; vascular tissues; and apical buds.	[6,107]
*K. ceratophylla*	Perennial, succulent, and glabrous species.		[16]
*K. laciniata*	Perennial or biennial herb that grows from 30 cm to 1.5 m in height. Its leaves are oval, opposite, fleshy, simple, short-petiolate, glossy, and pale green to dark green in color. They have dentate to crenate leaf margins, with a cylindrical herbaceous stem and fleshy petiole.	The secretory structures found in the stems, petioles and leaf blades consist of idioblasts that contain anthocyanins. The epidermis of *K. laciniata* is a single layer with adhering and oblong cells. The outer cell wall is convex and covered with cuticles. The leaves are amphistomatic and the chlorenchyma tissue is uniform. The cells of the chlorenchyma tissue have irregular, spherical-ellipsoidal shapes. The vacuoles of some mesophyll cells located near the epidermis, vascular bundles, and hydathodes contain phenolic compounds. The leaves show the presence of adaptive traits that enable them to survive in dry environments	[42,44,108]
*K. laxiflora*	Perennial species with multicolored leaves, that are crenate, green in shady settings, and pink or purple in bright sun. The flower buds are almost transparent but when they open, they turn orange.		[6]
*K. marmorata*	The leaves are large, oval, blue-green colored, with purple markings, arranged in stacked, opposite pairs to a height of 30 cm. The brown spots become brighter during summer dormancy and in strong sunlight; during winter they become pale or disappear altogether.		[3,6]
*K. orgyalis*	It is a much-branched slow-growing shrub that can reach approximately 1–2 m in height. It has spoon-shaped leaves, which are bronze to gray on the underside, and felted on the top of each leaf, with cinnamon-toned fuzz. Late winter or early spring brings bright yellow flowers in terminal clusters at the branch tips.		[6]
*K. pinnata*	An erect, succulent, perennial and glabrous plant that grows up to 1.5 m in height. The species reproduces through seeds and from leaf bulbils. The freshly dark green leaves are large (12–18 cm and 6–8 cm in size), simple, opposite, ovate, or elliptic, have serrate-crenate margins with buds, an obtuse apex, asymmetric base, reticulate venation, and long petiole. The flowers are pendulous, dark, and bell-like. The stems are tall, hollow, obtuse, and four-angled. The fruits are enclosed in the calyx and corolla. The seeds are small, smooth, oblong-ellipsoid, rarely striate, and smooth.	The leaves are broadly shallow on the adaxial side and convex on the abaxial side. The epidermal layer is thin, with small prominent cells on the adaxial side and less distinct on the abaxial side. The ground tissue of the midrib is parenchymatous and homogenous. The cells are circular or angular and compact. The vascular strand is single, collateral, small, and hemispherical; it consists of a thick horizontal band of xylem and a wide band of phloem. The lamina is uniformly flat with an even surface. The mesophyll tissue is not differentiated into palisade and spongy parenchyma. The stomata are anisocytic. The leaf petiole shows prismatic crystals of calcium oxalate embedded in parenchymatous cells, and annular and spiral vessels. In the powder, part of the vascular bundle, epidermis, annular and spiral xylem vessels were observed. The secretory structures found in the stems, petioles, and leaf blades consisted of idioblasts containing anthocyanins.	[12,42,44,63,78,79,106,109]
*K. pumila*	It is a 30 cm high shrublet with small, fleshy leaves covered with powdery deposits formed by calcium carbonate sediments. The leaves are obovate (2.8 cm long, 1.7 cm wide, and 2.5 mm thick), opposite, wedge-shaped, and have a sinuate basis and dentate-serrate margins.	The reddish-brown or purple color appears along the leaf margins after exposure to sunlight due to the presence of anthocyanins in the epidermal cells and mesophyll vacuoles. The epidermal cells are polygonal–isodiametric or slightly oblong; they are more numerous on the abaxial surface. The anticlinal walls are curved or straight and are convex on the outer walls. The walls are thickened due to the presence of wax. The cuticula is smooth or slightly undulating, elevated or with striae, with sparse white or gray irregularly shaped and sized wax structures on the surface. The leaves are amphistomatic, with anisocytic stomata. The vascular bundles are collateral and closed. The sheath cells, or phloem, xylem parenchyma cells, subepidermal ground tissue, mesophyll tissue, and chlorenchyma tissue cells may contain tannin substances.	[110]
*K. rhombopilosa*	Small plant (no more than 10 cm tall), which blooms in spring. The leaves are hard and triangular, with a pale and wavy margin and green-yellow flowers with red lines.		[6]
*K. synsepala*	One of the more unusual species of the genus because it is one of the few that produces stolon (lateral spreading stems). The leaves are arranged in rosettes and are thick, succulent, smooth, shiny, and green, with violet-red marks along the margins. This species is dormant in winter. The flowers are small, hairy, tubular, numerous, and pink.		[6]
*K. tetraphylla*	The leaves are silvery pale green, which turn red in bright sun and revert to green in active growth. It has a large rosette of rounded or wavy leaves. The inflorescence is terminal and erect, with densely clustered panicles of greenish, waxy, narrow, urn-shaped flowers.		[6]
*K. tomentosa*	The leaves are silvery, about 30 cm tall, reflecting the sun’s rays, lessening the chances of leaves overheating.	Its dense trichomes arise in triplets and perform a vital function in dry environments, helping to reduce the transpiration of water from the leaf surface	[6]
*K.* × *houghtonii*	A perennial erect herb, monocarpic, and can reach a height of up to 1.5 m. The leaves are opposite or verticillate, petiolate, with the leaf blade simple. The leaves vary from triangular to narrowly lanceolate, are serrate and mottled. The species forms corymbiform inflorescences of more than 100 pendulous, tetra or pentameric, dark-red flowers.		[102]

These data demonstrate that even with some similarities between the species, an adequate morpho-anatomical study of the material can allow the correct identification of the studied species and validation of the scientific data (biological or chemical study). In this review, species identification errors that could disavow the scientific data obtained have been identified [29,106]. 

### 2.3. Chemical Composition

There have been 124 chemical metabolites reportedly isolated from *Kalanchoe* species (Table 4, Table 5 and Table 6 and Figure 3, Figure 4 and Figure 5). The most common are cardiac glycosides (compounds **1**–**39**, in Table 4 and Figure 3) and flavonoids (compounds **40**–**78**, in Table 5 and Figure 4). The primary species from which these compounds have been reported are *K. daigremontiana*, *K. pinnata*, *K. delagoensis*, and *K. ceratophylla*. Cardiac glycosides (such as the bufadienolide class) have been identified in the species *K. ceratophylla*, *K. daigremontiana*, *K. delagoensis*, *K. hybrida*, *K. lanceolata*, and *K. pinnata* (compounds **1**–**39**, in Table 4 and Figure 3).

**Table 4 molecules-28-05574-t004:** Cardiac glycosides from *Kalanchoe* species.

Extract and/or Plant Part	Compound Name	Species	References
Dichloromethane extract; methanol extract of aerial parts; flowers	bryophyllin A (bryotoxin C) (**1**)	*K. daigremontiana*; *K. pinnata*; *K. delagoensis*; *K. ceratophylla*	[11,32,86,111,112,113,114,115]
Aqueous extract from the roots or whole plant; methanol extract of aerial parts	bryophyllin B (**2**)	*K. daigremontiana*; *K. ceratophylla*; *K. delagoensis*; *K. pinnata*	[30,111,113,114,116]
Leaves, dichloromethane extract; methanol extract of the leaves; dichloromethane fraction from methanol extract	bryophyllin C (**3**) daigremontianin (**4**) methyl daigremonate (**5**)	*K. daigremontiana*; *K. pinnata*	[11,112,117]
Dichloromethane extract; aqueous extract from the roots	bersaldegenin-1,3,5-orthoacetate (**6**)	*K. daigremontiana*; *K. pinnata*; *K. delagoensis*	[11,30,32,86,116]
Aqueous extract from the roots; ethanol and dichloromethane extracts from the roots; leaves	bersaldegenin 1-acetate (**7**)	*K. daigremontiana*; *K. pinnata*; *K. delagoensis*	[11,30,32,86]
Leaves; ethanol and dichloromethane extracts from the leaves	bersaldegenin 3-acetate (**8**)	*K. pinnata*	[11,86,114]
Aqueous extract from the roots	bersaldegenin (**9**)	*K. daigremontiana*	[30]
Ethanol and dichloromethane extracts from the leaves	bufalin (**10**)	*K. pinnata*	[86]
Flower heads	bryotoxin A (**11**)	*K. delagoensis*	[111]
Aqueous extract from the roots; flowers	bryotoxin B (**12**)	*K. daigremontiana*; *K. delagoensis*; *K. pinnata*	[30,111,115,116]
Aqueous leaf extract; leaves	bufadienolide A (**13**) bufadienolide B (**14**)	*K. daigremontiana*	[118]
Aqueous extract from the roots	daigredorigenin 3-acetate (**15**)	*K. daigremontiana*	[30,116]
	11α,19-dihydroxytelocinobufagin (**16**)		
Methanol extract of aerial parts	hellebrigenin (**17**)	*K. ceratophylla*	[113]
Methanol extract of aerial parts	hellebrigenin-3-acetate (**18**)	*K. ceratophylla*; *K. daigremontiana*	[113,116]
Methanol extract of aerial parts	kalanchoside A (**19**) kalanchoside B (**20**) kalanchoside C (**21**)	*K. ceratophylla*	[113]
Methanol extract of aerial parts	thesiuside (**22**)	*K. ceratophylla*	[113]
Ethanol extract; whole plant	kalantuboside A (**23**) kalantuboside B (**24**)	*K. delagoensis*	[32]
Aqueous extract from the roots	1β,3β,5β,14β,19-pentahydroxybufa-20,22-dienolide (kalandaigremoside A) (**25**) 19-(acetyloxy)-1β,3β,5β,14β-tetrahydroxybufa-20,22-dienolide (kalandaigremoside B) (**26**) 3β-(O-α-L-rhamnopyranosyl)-5β,11α,14β,19-tetrahydroxybufa-20,22-dienolide (kalandaigremoside C) (**27**) 19-(acetyloxy)-3β,5β,11α,14β-tetrahydroxybufa-20,22-dienolide (kalandaigremoside D) (**28**) 3β,5β,11α,14β,19-pentahydroxy-12-oxo-bufa-20,22-dienolide (kalandaigremoside E) (**29**) 19-(acetyloxy)-3β,5β,11α,14β-tetrahydroxy-12-oxo-bufa-20,22-dienolide (kalandaigremoside F) (**30**) 19-(acetyloxy)-1β,3β,5β,11α,14β-pentahydroxy-12-oxo-bufa-20,22-dienolide (kalandaigremoside G) (**31**) 1β-(acetyloxy)-3β,5β,11α,14β,19-pentahydroxy-12-oxo-bufa-20,22-dienolide (kalandaigremoside H) (**32**)	*K. daigremontiana*	[30]
Ethyl acetate extract of the fresh; whole plant	lanceotoxin A (**33**) lanceotoxin B (**34**)	*K. lanceolata*	[119]
Methanol extract; whole plant	kalanhybrin A (**35**) kalanhybrin B (**36**) kalanhybrin C (**37**)	*K. hybrida*	[120]
Ethanol extract of the whole plant	kalantubolide A (**38**) kalantubolide B (**39**)	*K. delagoensis*	[32]

Flavonoids have been identified in aqueous, hydroalcoholic, and alcoholic extracts from the leaves of *K. blossfeldiana*, *K. crenata*; *K. daigremontiana*, *K. delagoensis*, *K. fedtschenkoi*, *K. laciniata*, *K. marmorata*, *K. mortagei*, and *K. pinnata* (compounds **40**–**78**, Table 5 and Figure 4). The most common flavonoids/glycosylated flavonoids described from these species are derivatives of quercetin (**40**), patuletin (**69**–**71**), eupafolin (**64**), and kaempferol (**48**–**62**). 

**Table 5 molecules-28-05574-t005:** Flavonoids from *Kalanchoe* species.

Extract and/or Plant Part	Compound Name	Species	References
Flower; ethanol leaf extractFlowers; Leaves	quercetin (**40**)	*K. pinnata* *K. delagoensis* *K. blossfeldiana*; *K. mortagei*; *K. fedtschenkoi*; *K. daigremontiana*; *K. longiflora* *K. ceratophylla*	[32,37,121,122,123,124,125,126,127]
Flower extractFlowers	Quercetin 3-*O*-β-glucoside (quercetin 3-*O*-glucoside; isoquercetin; isoquercetrin) (**41**)	*K. pinnata*; *K. blossfeldiana*; *K. daigremontiana*; *K. delagoensis*	[49,122,123,127,128]
Flower extractFlowers	quercetin 3-*O*-β-d-glucuronopyranoside (miquelianin) (**42**)	*K. pinnata*	[122]
Aqueous and methanolic leaf extractsLeaves	quercetin 3-*O*-rhamnoside (quercitrin) (**43**)	*K. pinnata*; *K. delagoensis*; *K. longiflora*; *K. ceratophylla*	[42,82,122,123,125,126,129,130]
Flowers, Aqueous leaf extractFlower, Leaves	quercetin-3-*O*-β-d-xylopyranosyl (1→2)-α-L-rhamnopyranoside (**44**)	*K. blossfeldiana* *K. daigremontiana*	[118,127]
Aqueous and methanolic leaf extracts; flower extractFlowers, Leaves	quercetin 3-*O*-α-l-arabinopyranosyl-(1→2)-α-l-rhamnopyranoside (**45**)	*K. pinnata*	[8,82,83,122,129,130]
Methanol leaf extractLeaves	quercetin 3-*O*-α-l-arabinopyranosyl-(1→2)-α-l-rhamnopyranoside-7-*O*-β-d-glucopyranoside (**46**)	*K. pinnata*	[129]
Ethanol leaf extractLeaves	quercetin 3-*O*-rutinoside (rutin) (**47**)	*K. pinnata*	[121]
Methanolic and hydroethanolic extracts from the leavesLeaves	kaempferol (**48**)	*K. delagoensis*; *K. pinnata*; *K. fedtschenkoi*; *K. longiflora*; *K. ceratophylla*	[2,37,43,123,125,126,129,131]
Water and ethanol extracts Leaves	kaempferol 3,7-*O*-dirhamnoside (kaempferitrin) (**49**) kaempferol 3-*O*-β-d-xylopyranosyl-(1→2)-α-L-rhamnopyranoside-7-*O*-β-d-glucopyranoside (daigremontrioside) (**50**)	*K. daigremontiana*	[49]
Leaves	kaempferol 7-*O*-rhamnoside (**51**)	*K. delagoensis*; *K. longiflora*	[123,125]
Methanol leaf extractLeaves	kaempferol 3-*O*-β-d-xylopyranosyl-(1→2)-α-l-rhamnopyranoside (kaempferol 3-*O*-xylosyl-rhamnoside) (**52**)	*K. pinnata*; *K. daigremontiana*	[49,118,129]
Leaves	kaempferol 3-*O*-galactoside (trifolin) (**53**)	*K. delagoensis*	[123]
Leaves	kaempferol 3-rutinoside (nicotiflorin) (**54**)	*K. pinnata*; *K. longiflora*	[70,125]
Leaves	kaempferol- 3-*O*-robinoside-7-*O*- rhamnoside (robinin) (**55**)	*K. delagoensis*; *K. longiflora*	[123,125]
Aqueous and methanolic leaf extractsLeaves	kaempferol 3-*O*-α-l-arabinopyranosyl (1→2)-α-l-rhamnopyranoside (kapinnatoside) (**56**)	*K. pinnata*	[83,129,130]
Ethyl acetate extract of the wholeWhole plant	kaempferol 3-*O*-α-l-(2-*O*-acetyl)rhamnopyranoside 7-*O*-α-l-rhamnopyranoside (**57**) kaempferol 3-*O*-α-l-(3-*O*-acetyl)rhamnopyranoside 7-*O*-α-l-rhamnopyranoside (**58**) kaempferol 3-*O*-α-l-(4-*O*-acetyl)rhamnopyranoside 7-*O*-α-l-rhamnopyranoside (**59**) kaempferol 3-*O*-α-d-glucopyranoside 7-*O*-α-l-rhamnopyranoside (**60**) afzelin (kaempferol 3-*O*-α-l-rhamnopyranoside) (**61**) α-rhamnoisorobin (kaempferol 7-*O*-α-l-rhamnopyranoside) (**62**)	*K. pinnata*	[132]
Aqueous leaf extractLeaves	4′,5-dihydroxy-3′,8-dimethoxyflavone 7-*O*-β-d-glucopyranoside (**63**)	*K. pinnata*	[130]
Aerial parts; methanol extract from the stemsStems	eupafolin (6-methoxyluteolin) (**64**)	*K. ceratophylla*	[17,126]
Aerial parts	eupafolin 4′-*O*-rhamnoside (**65**)	*K. ceratophylla*	[126]
Ethanol extract of the wholeWhole plant	4′-methoxyherbacetin (**66**)	*K. delagoensis*	[32]
Stems and leaves; Leaves	kalambroside A (**67**) kalambroside B (**68**) kalambroside C (**69**) patuletin 3-*O*-(4′-*O*-acetyl-α-l-rhamnopyranosyl)-7-*O*-(3′-*O*-acetyl-α-l-rhaminopyranoside) (**70**) patuletin 3-*O*-α-l-rhamnopyranosyl-7-*O*-(3′-*O*-acetyl-α-L-rhaminopyranoside) (**71**)	*K. laciniata*	[133]
Stems and leaves; hydroethanolic extract from leavesStems; Leaves	patuletin 3-*O*-α-l-rhamnopyranosyl-7-*O*-α-l-rhamnopyranoside (**72**)	*K. laciniata*	[40,44,133]
Methanol leaf extract Leaves	myricetin 3-*O*-α-l-arabinopyranosyl-(1→2)-α-l-rhamnopyranoside (**73**) myricitrin (myricetin 3-*O*-α-l-rhamnopyranoside) (**74**) diosmine (diosmetin 7-*O*-α-l-rhamnopyranosyl-(1→6)-β-d-glucopyranoside) (**75**) acacetin 7-*O*-α-l-rhamnopyranosyl-(1→6)-β-d-glucopyranoside (**76**)	*K. pinnata*	[129]
Ethanol leaf extractLeaves	luteolin (**77**)	*K. ceratophylla*; *K. pinnata*	[121,126]
Ethanol leaf extractLeaves	luteolin 7-*O*-β-d-glucoside (**78**)	*K. pinnata*	[121]

Recently, a comprehensive approach encompassing metabolomics and machine learning techniques was implemented [134] to investigate *K. daigremontiana*, *K.* × *houghtonii*, and *K. delagoensis* plant tissue cultures. By employing untargeted metabolomics, a remarkable total of 460 phenolic compounds were identified. Among them, the elicitation process significantly influenced the biosynthesis of 164 compounds. Through the utilization of neuro fuzzy logic, the study successfully predicted the impact and interactions involved in plant cell growth as well as the biosynthesis of various subfamilies of polyphenols. The findings highlight the distinct genotype-dependent role of salicylic acid in eliciting *Kalanchoe* cell cultures, while methyl jasmonate emerged as a secondary contributing factor.

Several other secondary metabolites (steroids, triterpenes, coumarins, and others) have also been isolated from different species of *Kalanchoe* and are described in the literature (compounds **79**–**124**, in Table 6 and Figure 5).

**Table 6 molecules-28-05574-t006:** Other compounds isolated and identified from *Kalanchoe* species.

Class	Extract and/or Plant Part	Compound Name	Species	References
Aurone	Aqueous root extractRoots	hovetrichoside C (**79**)	*K. daigremontiana*	[116]
Coumarin	Aerial parts	7-hydroxycoumarin (**80**)	*K. ceratophylla*	[126]
Glycoside	Roots	KPB 100 (**81**) KPB 200 (**82**) schisandriside (**83**)	*K. pinnata* *K. daigremontiana*	[69,116]
Glycoside	Aqueous root extract	schisandriside (**83**)	*K. daigremontiana*	[116]
Lipid	Ethanol extract of the wholeWhole plant	taurolipid C (**84**)	*K. delagoensis*	[32]
Megastigmane	Ethanol extract of the wholeWhole plant	(6S,7R,8R,9S)-6- oxaspiro-7,8-dihydroxymegastigman-4-en-3-one (tubiflorone) (**85**)	*K. delagoensis*	[32]
Organic/ phenolichenolic acid	Leaves, ether leaves extract	ferulic acid (**86**)	*K. delagoensis*; *Kalanchoe* sp. *K.daigremontiana* *K. pinnata*;	[29,123,135,136]
	Ethanol extract of the wholeWhole plant; leavesLeaves	gallic acid (**87**)	*K. delagoensis*; *Kalanchoe* sp.; *K. daigremontiana*	[29,32,123,135]
	Leaves; ether leaves extract	caffeic acid (**88**)	*K. delagoensis*; *Kalanchoe* sp.; *K. longiflora*; *K. daigremontiana* *K. pinnata*	[29,123,125,135,136]
	Leaves	protocatechuic acid (**89**)	*K. delagoensis*; *Kalanchoe* sp.; *K. daigremontiana*	[29,123,135]
	Ethanol extract of the whole plant; leaves; ether leaves extractWhole plant; Leaves	syringic acid (**90**)	*K. delagoensis*; *Kalanchoe* spp.; *K. pinnata*	[32,123,135]
	Leaves	sinapic acid **91**)	*Kalanchoe* sp.	[135]
	Ethanol extract of the wholeWhole plant; leavesLeaves	vanillic acid (**92**)	*K. delagoensis*; *Kalanchoe* sp.	[32,135,136]
	Leaves	chlorogenic acid (**93**)	*Kalanchoe* sp.; *K. longiflora*	[125,135]
	Leaves; ether leaves extract	p-Coumaric acid (**94**)	*Kalanchoe* sp.; *K. longiflora*; *K. daigremontiana* *K. pinnata*	[29,125,135,136]
	Leaves	β- resorcylic acid (**95**)	*Kalanchoe* sp.	[135]
		γ-resorcylic acid (**96**)		
	Ethanol extract of the wholeWhole plant	cinnamic acid (**97**) 4-*O*-ethylgallic acid (**98**) methyl gallate (**99**)	*K. delagoensis*	[32]
Phenolic compounds	Whole plant	4-*O*-ethylgallic acid (**98**) methyl gallate (**99**) 3,4-dimethoxyphenol (**100**) phloroglucinol (**101**) 3,4-dihydroxyallylbenzene (**102**)	*K. delagoensis*	[32]
Phenanthrene	Leaves	bryophollenone (**103**) 2(9-decenyl) phenanthrene (**104**)	*K. pinnata*	[137]
Steroid	Leaves	bryophyllol (**105**) 24-ethyl-25-hydroxycholesterol (**106**)	*K. pinnata*	[137]
		24-ethyl-25-hydroxycholesterol (**106**)		
	Ethanol extract of the wholeWhole plant	stigmasterol-*O*-d-glucoside (**107**)	*K.delagoensis*	[32]
Tocochromanol	Hexane leaf extractLeaves	α-tocopherol (**108**) γ-tocopherol (**109**) δ-tocopherol (**110**) β-tocomonoenol (**111**) γ-tocomonoenol (**112**) δ-tocomonoenol (**113**)	*K. daigremontiana*	[138]
iterpeneTriterpene	Aerial parts; petroleum ether extract from flowers; methanol extract Flowers	friedelin (**114**)	*K. fedtschenkoi*; *K. marnieriana*; *K. daigremontiana* *K. integra*	[136,139,140]
		glutinone (**115**)	*K. miniata*	[139]
		glut-5-en-3- β-ol (glutinol) (**116**)	*K. fedtschenkoi*; *K. daigremontiana* *K. integra*	[136,139,140,141]
	Leaves	18α-oleanane (**117**) α-amyrin acetate (**118**)	*K. pinnata*	[137]
		α-amyrin acetate (**118**)		
	Leaves	β-amyrin acetate (**119**)	*K. pinnata*; *K. miniata*	[137,139]
	Leaves; methanol extract	α-amyrin (**120**)	*K. pinnata* *K. daigremontiana*	[137,141]
	Leaves; methanol extract	β-amyrin (**121**)	*K. pinnata*;	[137,140,141]
	Leaves, petroleum ether extract from flowers; Flowers	bryophynol (**122**) Ψ-taraxasterol (**123**) bryophollone (**124**)	*K. daigremontiana* *K. pinnata* *K. integra*	[136,137]

Until now, of the four species most reported as medicinal plants with ethnopharmacological use (*K. pinnata*, *K. laciniata*, *K. crenata*, and *K. daigremontiana*), only two had cardiac glycosides identified in published studies (*K. pinnata* and *K. daigremontiana*). In contrast, compounds from the flavonoid class were identified in all four species. Additionally, although the juice or crude extract (produced by maceration) is the ethnomedicinal form of use in the literature, phytochemical studies are generally based on polar organic extracts (ethanol, methanol) prepared from leaves, stems, roots, flowers, and whole plant. Few studies using nonpolar or aqueous solvents have been identified. This is an important observation because it is known that popular knowledge needs to be confirmed, and the presence of the biological compounds in an extract are related to the solvent and the procedure used to obtain it.

### 2.4. Pharmacological Activities

In folk medicine, the use of *Kalanchoe* species is related to several disease conditions. Due to its widely distributed and popular use, experiments have been performed to corroborate the pharmacological potential activities and to prove the therapeutic potential of different species of *Kalanchoe*. So far, only 16 of the 133 species of the genus *Kalanchoe* have been analyzed to assess various pharmacological activities. The primary activities studied have been antioxidant, anti-inflammatory, cytotoxic, and antimicrobial properties. Of these sixteen species, four are not reported in the literature regarding their popular uses, but their pharmacological activities were tested (*K. blossfeldiana*, *K. longiflora*, *K. scapigera*, and *K. rhombopilosa*).

*Kalanchoe blossfeldiana* methanolic extract (ME) showed biofilm formation and demonstrated anticytokine properties [128]. Its aqueous extract (AE) in zinc oxide nanoparticles showed promising antibacterial and antifungal potential and a potent cytotoxic effect against a HeLa cell line [142]. In comparison with two other species (*K. daigremontiana* and *K. pinnata*), the ethanolic extract (EE) of *K. blossfeldiana* exhibited the most potent cytotoxic activity (IC_50_: < 19 µg/mL for HeLa and SKOV-3 cells) and the strongest antibacterial effects (MIC: 8.45, 8.45, 0.25, and <33.75 µg/mL for *S. aureus*, *S. epidermidis*, and *E. hirae*, respectively) but this extract did not contain bufadienolides, which are known to elicit these biological effects (cytotoxic and antibacterial) [11].

*Kalanchoe ceratophylla* stems ME has been suggested to provide analgesic and anti-inflammatory effects, with its anti-inflammatory mechanisms being well discussed. Eupafolin (**64**) demonstrated good pharmacological activity, and the antioxidant potential and efficacy of this species may be largely attributed to polyphenolic compounds [16,17]. The antiviral effects of the leaf extract from this species were investigated against RNA enteroviruses, specifically enterovirus 71 (EV71) and coxsackievirus A16 (CVA16). The extract showed little cytotoxicity and exhibited concentration-dependent antiviral activities, including reductions in cytopathic effects, plaque formation, and virus yield. Furthermore, the extract demonstrated greater potency in antiviral activity compared to ferulic acid, quercetin, and kaempferol, significantly inhibiting the in vitro replication of EV71 (IC_50_: 35.88 μg/mL) and CVA16 (IC_50_: 42.91 μg/mL). As such, this extract may be considered a safe anti enteroviral agent [143].

*Kalanchoe crenata* ME was non-toxic when administered orally for animals over a period of 14 days. The ME and its fractions showed fold decreases in IC_50_ for fractions regarding CYP3A4; phytoconstituents in the ME were a reversible and time-dependent inhibitor of CYP3A4, and the methanol fraction is a potential source of a new oral anti-nephropathic drug [18,20,22]. The cytotoxicity of ME leaves was highlighted in comparison with five other species, with reported IC_50_ values that ranged from 2.33 μg/mL (SPC212, mesothelioma) to 28.96 μg/mL (HepG2, hepatocarcinoma), and apoptosis induction via ROS production [23]. Its AE were quantitatively assessed for significant elements, and the amounts of Ca, K, and Mg detected could be correlated to its traditional usage in cases of hypertension and arrhythmia. However, the presence of heavy metals (Pb and As inorganic) may be a major health concern [39]. The AE antidepressant potential could be possibly mediated by a complex interplay between serotoninergic, opioidergic, and noradrenergic systems [75]. The EE showed no genotoxic potential and possessed cardioprotective effects against DOX-induced cardiotoxicity in Sprague-Dawley rats [19]. The methylene chloride/methanol extract and its hexane, methylene chloride, ethyl acetate, n-butanol fractions, and aqueous residue were evaluated for their analgesic effects and anticonvulsant activity. The results suggested the presence of peripheral and central analgesic activities, along with an anticonvulsant effect [144].

Bufadienolide-rich fractions (BRF) isolated from the roots of *K. daigremontiana* presented antioxidant activity against DPPH radicals (EC_50_: 21.80 µg/mL); moderate activity for peroxynitrite-induced oxidative stress; protective levels of 3-nitrotyrosine and thiol groups (50 µg/mL); effective antioxidant potential for hydroperoxides and TBARS generation (1–5 and 25–50 µg/mL, respectively); uncompetitive inhibitory effect on the enzymatic properties of a serine proteinase-thrombin (1–50 µg/mL) (IC_50_: 2.79 µg/mL); and of plasmin (0.05–50 µg/mL). No effects were observed to prevent the oxidation of low-molecular plasma thiols, and no cytotoxicity was observed. Docking studies suggested that only some compounds (mostly bersaldegenin 1-acetate (**7**), bryotoxins (**1**,**11**–**12**), and hovetrichoside C (**78**)) were bound to plasminogen/plasmin, depending on the presence or absence of the substrate in the active site, suggesting allosteric regulation of plasminogen activation and plasmin activity by components of the examined fraction [14,15,116]. Additionally, root extracts of *K. daigremontiana* was also evaluated [145] in comparison to other plants (*Cyphomandra betacea*, *Robinia pseudoacacia*, *Nothofagus pumilio*, and *Rosmarinus officinalis*) in a set of in vitro assays and, regarding the cytotoxic assays, *K. daigremontiana* was the only species considered to be highly toxic. 

The anti-inflammatory activity of AE, EE, and petroleum ether (PEE) extracts obtained from the leaves of *K. pinnata* and *K. daigremontiana* were compared and the AE and PEE of *K. daigremontiana* showed the highest anti-inflammatory effects (−105.69 ± 0.40 and −79.95 ± 0.37, respectively) [106]. Crude extracts from the leaves of *K. daigremontiana* can contribute to antiviral activity [118] and, most prominently, to high antibacterial activity [10] against *E. coli* and *S. aureus*. A macerated ME from the leaves of *K. daigremontiana* demonstrated high antiparasitic activity against *E. histolytica* and *T. vaginalis* (IC_50_: 70.71 ± 3.08 and 105.27 ± 5.19 μg/mL, respectively) [124]. Antioxidant properties of nanovesicle preparations of *K. daigremontiana* compared to *Artemisia absinthium*, *Hypericum perforatum*, *Silybum marianum*, *Chelidonium majus*, and *Scutellaria baicalensis* demonstrated that the activities are specific to plant species, but *K. daigremontiana* and *S. marianum* nanoparticle showed similar characteristics, suggesting future analysis to test the complementary/synergic effects between them [146].

The cytotoxic effects of *K. daigremontiana* were investigated in relation to human adenocarcinoma (HeLa), ovarian (SKOV-3), breast (MCF-7) and melanoma (A375) cells [49,147], and human multiple myeloma cells [28]. The dichloromethane fraction (DF) showed strong activity against all cell lines (IC_50_ ≤ 10 µg/mL), and it could be related to the presence of bersaldegenin-1,3,5-orthoacetate (**6**). The AE reduced the viability of tumor cells by 13% and, in combination with doxorubicin, showed an additive synergism of action, which enhanced this effect. The intracellular glutathione level decreased by 25%, mitochondrial membrane potential decreased by 19%, and ATPase activity increased 50%, which shows that this extract affects the metabolism of tumor cells and contributed to their death and antitumor activity. The AE elevated the oxidative stress levels in SKOV-3 cells as well as exhibited notable antiproliferative and cytotoxic effects, leading to the depolarization of the mitochondrial membrane and causing a significant cell cycle arrest in the S and G2/M phases of this cell line. The non-activation of caspases 3, 7, 8, and 9 suggests a non-apoptotic mode of cell death. Additionally, real-time PCR analysis suggested that the AE may induce cell death through the involvement of TNF receptor (tumor necrosis factor receptor) superfamily members 6 and 10.

The *K. delagoensis* n-hexane and ethanol extracts suggested wound-healing potential [34]. Its n-butanol-soluble fraction was able to inhibit cell proliferation and reduce cell viability by two mechanisms exclusively involved with cell division (inducing multipolarity and disrupting chromosome alignment during metaphase) [31]. The AE of this species promoted cell cycle arrest and senescence-inducing activities in A549 cells, and tumor growth was effectively inhibited, suggesting that this extract is an antitumor agent [148]. Compounds isolated from the EE of this species were evaluated for anti-inflammatory and cytotoxic activities [32,33]. Some compounds (quercetin (**40**), syringic acid (**84**), 3,4-dimethoxyphenol (**94**), 3,4-dihydroxyallylbenzene (**96**), and tubiflorone (**120**)) possessed NO inhibitory activity (IC_50_ 15.1/0.9–98.9/1.3 mM). The biological evaluation indicated that some cardenolides (kalantubolide A (**38**) and kalantubolide B (**39**)) and bufadienolide glycosides (bryophyllin A (**1**), bersaldegenin-1,3,5-orthoacetate (**6**), bersaldegenin 1-acetate (**7**), kalantuboside A (**23**), kalantuboside B (**24**)) demonstrated strong cytotoxicity against four human tumor cell lines (A549, Cal-27, A2058, and HL-60) (IC_50_ 0.01–10.66 µM). In addition, these compounds blocked the cell cycle in the G2/M-phase and induced apoptosis in HL-60 cells.

The ethyl acetate extract (EAE) of *K. flammea* is non-genotoxic and exhibits selective cytotoxic activity against several cell lines of prostate cancer, with mechanisms of induced apoptosis by the intrinsic pathway, significant downregulation of apoptosis-related proteins, induced DNA fragmentation, and cell cycle arrest. Additionally, a fraction rich in coumaric acid and palmitic acid, obtained from the EAE, demonstrated selective cytotoxic activity against PC-3 cells [36]. Similarly, fraction rich in fatty acids obtained from the EE of *K. pinnata* demonstrated inhibited lymphocyte proliferation in vitro and showed in vivo immunosuppressive activity [149].

*Kalanchoe fedtschenkoi* and *K. mortagei* were studied to compare their antibacterial potential [37], and *K. fedtschenkoi* extracts demonstrated growth inhibitory effects against *A. baumannii*, *P. aeruginosa*, and *S. aureus*, and its stem extracts exhibited the best inhibitory activity against *A. baumannii* (IC_50_ 128 µg/mL). Four treatments (250 µg/mL for 72 h) with different parts of the AE of *K. gastonis-bonnieri* inhibited the proliferation of benign prostatic hyperplasia (BPH) cells (13.5–56.7%), and the AE of underground parts was the most active, stimulating changes in the BPH cells and modulating crucial processes such as proliferation, viability, and apoptosis [38].

In a study that compared 57 extracts obtained from 18 plants, *K. glaucescens* possessed the second-highest antioxidant activity and considerable cytotoxicity against leukemia cells [150]. The *K. laciniata* extracts from leaves picked before and during blooming (BB and DB, respectively) were tested to assess anti-inflammatory effects and both extracts presented no acute toxicity in mice (0.25 to 5 g/kg). Oral doses of the BB (0.25, 0.5, and 1.0 g/kg) significantly inhibited paw edema during the first four hours after injection of 2% carrageenan but oral doses of the DB (0.5, 1.0 and 2.0 g/kg) had no inhibitory activity [81]. The AE of *K. laciniata* also displayed thyroid peroxidase inhibition [151], immunomodulatory and anti-inflammatory properties [152,153]. The aqueous-methanol (AM) and n-hexane (NH) extracts of this species showed significant mutagenicity and cytotoxicity, and the NH extract treatment was more sensitive than others to *E. coli* [47,48]. Hydroethanolic extracts (HEE) obtained from *K. laciniata* leaves indicated dose-dependent cytotoxic activity against a 3T3 cell line (normal) and the 786–0 line (kidney carcinoma) (92.23% cell inhibition). In an in vivo experiment, the extract showed only liver changes and damage related to acute toxicity, and no significant toxicity. The HEE was able to reduce *Salmonella* growth rate, and the cell number was reduced with the release of the bacterial content. This species is confirmed as a natural source of antioxidant agents [45,154].

The gastroprotective activity of the leaf juices of *K. laciniata* was evaluated and compared with *K. pinnata*, and both species showed gastroprotective effects; however, the *K. laciniata* extract reduced the lesions in all the tested doses [43]. Other authors [155] determined the effect of aqueous, ethanol, and hexane extracts of *K. laciniata* leaves in comparison to other plants (*Drymoglossum piloselloides* leaves and *Aegle marmelos* flowers) against CaOx urolithiasis in vitro and the results clearly demonstrated that all species have the capacity to inhibit the nucleation, growth, and aggregation of CaOx crystals. Preliminary phytochemical screening also revealed the presence of reducing sugars, proteins, flavonoids, tannins, and polyphenol compounds in *K. laciniata*. 

*Kalanchoe longiflora* was evaluated and compared to eight species of *Kalanchoe* in relation to antitrypanosomal, antimalarial, antileishmanial, cytotoxic, and antimicrobial activities [125]. This study revealed that *K. longiflora* leaf extracts showed activity against *T. brucei* with an inhibition concentration of sample at 50% (IC_50_ 17.6 µg/mL). To determine the mechanism of action of *K. longiflora* extract as a potent anti-trypanosomal and cytotoxic agent, the authors investigated the ability to inhibit topoisomerase I enzyme and found the *K. longiflora* extract showed the best activity (IC_50_ 0.148 µg/mL). 

The antioxidant potential of various extracts of *K. pinnata* were evaluated and significant dose-dependent antioxidant activity was demonstrated in all of them. The antioxidant activities of the AE from the leaves improves the antioxidant potential in various organs (mainly the aorta), prevents adverse changes due to CCl4 intoxication in rats by pre-treatment (25 and 50 mg/kg b.w.), and the inhibits arginase II, as well as increasing antioxidant status in CCl4-intoxicated rats, which suggests a protection of the kidneys against CCl4-induced oxidative damage [59,63,65,83]. The EE from its stem/bark was evaluated by DPPH and exhibited high antioxidant activity (IC_50_ 37.28 µg/mL). In comparison with other extracts (AE and PEE) obtained from the leaves, the EE showed the greatest radical inhibitory effect by DPPH, reaching a maximum inhibitory effect of 49.5 ± 5.6% (2000 µg/mL) [106,156]. The antioxidant property of ME from leaves showed 69.77% of free radical inhibition (100 µg/mL) of DPPH [62].

The concentration of vitamin C in AE of two *Kalanchoe* species (*K. daigremontiana* and *K. pinnata*) was evaluated and compared [12]; the amount of vitamin C was highest for the AE of *K. pinnata* (81 mg/100 g). Four major flavonoids obtained from HEE of *K. pinnata* leaves were evaluated by xanthine oxidase (XO) inhibition and antioxidant activity (DPPH and ABTS). It was found that kaempferol and quercetin derivatives moderately inhibited XO, while only quercetin derivatives displayed average radical scavenging activity, suggesting that quercetin 3-O-α-L-arabinopyranosyl-(1→2)-α-L-rhamnopyranoside (**45**) can be indicated as a specific marker of this species [71]. 

The *K. pinnata* AE and quercetin (**40**) inhibited degranulation and cytokine production of bone marrow-derived mast cells following IgE/FcRI crosslinking in vitro: they decreased the development of airway hyperresponsiveness, airway inflammation, goblet cell metaplasia, and production of IL-5, IL-13, and TNF in vivo. In contrast, treatment with quercitrin (**43**) did not affect the tested parameters [42]. Additionally, the AE and quercitrin showed protective effects in fatal anaphylactic shock [157].

The antinociceptive, antiedematogenic, and anti-inflammatory potential as well as the possible mechanisms of action of the subcutaneous administration of *K. pinnata* AE of flowers, its ethyl acetate (EAF), and butanol (BF) fractions, and the main flavonoid (**45**) were investigated in a mouse model; the AE and its main flavonoid produced antinociceptive, antiedematogenic, and anti-inflammatory activities through COX inhibition and TNF-𝛼 reduction [8]. The flowers AE also are described as a rich source of T-suppressive flavonoids that may be therapeutically useful against inflammatory diseases [122]. The AE and the EE of *K. pinnata* leaves were found to be effective as hepatoprotective, and the AE was more effective [158]. The AE of the *K. pinnata* leaves were also examined [159] to investigate the ulcer healing properties and gastroprotective activity. The results indicate that treatment with the AE exhibited a higher inhibition percentage compared to pretreatment with an isolated quercetin derivative. This suggests that while the isolated flavonoid may possess gastroprotective activity, other compounds present in *K. pinnata* could potentially act synergistically to enhance its effect.

Studies comparing the anti-inflammatory [131] and the anti-ophidic [44] activities of *K. laciniata* and *K. pinnata* have been performed. The anti-inflammatory activity of topical formulations containing AE of both species showed good results; however, *K. laciniata* was most effective, with excellent results on the formulation containing a low concentration of its AE (5%). On the other hand, even though HE extracts from both species significantly reduced the hemorrhagic activity of *B. jararaca* venom in pre-treatment protocol, only *K. pinnata* was active in the post-treatment protocol and in the anti-edematogenic activity assay. It was also more active in the phospholipase test. Continuing the study, the authors conducted a study [160] to evaluate the healing properties and mechanism of action of the topical formulation of *K. pinnata*, which demonstrated the ability to stimulate the healing of skin wounds, leading to a reduction in wound area. Additionally, it exhibited a notable decrease in inflammatory infiltrate, as well as lowered levels of IL-1β and TNF-α. Moreover, the formulation induced angiogenesis by increasing the expression of VEGF, similar to the effects of Fibrinase. These findings highlight the significant potential of this formulation as a novel active ingredient in the development of pharmaceuticals for wound healing. The EE of *K. pinnata* also shows wound healing activity [161].

A method for targeting and identifying molecules with antimicrobial activity was implemented, which could potentially replace chemical preservatives in cosmetic applications [70]. An in vitro evaluation of the antimicrobial activity of different extracts (petroleum ether, chloroform, methanol, and aqueous) produced from *K. pinnata* roots was performed against *E. coli*, *S. aureus*, *P. aeruginosa*, and *C. albicans* [78], and the ME presented as an effective antibacterial, while none of the extracts showed activity against *C. albicans*. The EE of stem bark was tested against antimicrobial activity [156]; it inhibited the growth of microorganisms such as *B. cereus*, *E. coli*, *S. aureus*, *P. aeruginosa*, *K. pneumoniae*, and *A. niger*, while the extract was inactive against *S. typhi* and *C. albicans*. 

The AE of *K. pinnata* displayed a significant reduction in hepatic and splenic parasite burden, indicating that the oral efficacy of this species extends to visceral leishmaniasis caused by *L. chagasi* [73,162]. The antileishmanial activity of three flavonoid glycoside and free quercetin (**40**) (isolated from the AE of *K. pinnata*) were also demonstrated [82,130], with a low toxicity profile. The anthelmintic capacity of the PEE and ME of *K. pinnata* was explored [163]. Both extracts were investigated in different concentrations for anthelmintic activity against *P. posthuma* and they exhibited no anthelmintic activity even at the highest concentration (200 mg/mL); the conclusion was that they had no vermicide activity. Two compounds (KPB-100 (**122**) and KPB-200 (**123**)) identified from *K. pinnata* are promising targets for synthetic optimization and in vivo study against human alpha herpesvirus 1 and 2 and vaccinia virus. KPB-100 (**122**) inhibited all the tested viruses [69]. The bryophyllin A (**1**) (isolated from *K. pinnata*), bersaldegenin 1,3,5-orthoacetate (**6**) and daigremontianin (**4**) (isolated from *K.* x *houghtonni*) showed good inhibitory potential on the Epstein-Barr virus, but bryophyllin A (**1**) was the most effective (IC_50_: 0.4 µM) [112]. Additionally, both bryophyllin A and C isolated from a ME of the leaves of *K. pinnata* showed strong insecticidal activity against third instar larvae of the silkworm.

The EE of *K. pinnata* shows great hypoglycemic effect and the improvement of the number of pancreatic Langerhans beta cells at medium-dose treatment (11.6 mg/kg); it has a hypoglycemic effect through the improvement of the number of pancreatic Langerhans beta-cells. On the other hand, the DF from AE of *K. pinnata* demonstrates a dose-dependent insulin secretagogue action; reducing fasting blood glucose values (from 228 mg/dL to 116 mg/dL, on 10 mg/kg); improving the glycated hemoglobin to 8.4% (compared with 12.9% in diabetic controls); and restoring insulin level and lipid profile values close to normal [87,90]. The antioxidant effects of combined preparations of *K. pinnata* and metformin were investigated [164]. The treatment with *K. pinnata* alone (400 µg/mL), resulted in a significant increase in catalase activity in both non-diabetic and diabetic human skeletal muscle myoblasts, as well as in a human skeletal muscle myoblast cell line subjected to H_2_O_2_-stress-induced stress. Simultaneously, *K. pinnata* treatment led to a significant reduction in malondialdehyde levels. Notably, the combination of *K. pinnata* and metformin appeared to modulate antioxidant responses by increasing the enzymatic activity of superoxide dismutase, elevating the levels of reduced glutathione, and reducing glutathione levels in both non-diabetic and diabetic human skeletal muscle myoblasts, as well as in the H_2_O_2_-stress-induced human skeletal muscle myoblasts, which demonstrates the potential of these treatment in addressing the pathophysiological complications linked to oxidative stress in individuals with type II diabetes.

Investigations of the in vitro cytotoxicological and genotoxicological effects of *K. pinnata* were performed using its AE [85], leaf juice [64], and EE [165]. All the results indicated significantly lower results than those found for a positive control, suggesting a weak genotoxic response or a non-genotoxic effect. Hence, these extracts of *K. pinnata* can be used, but not for long durations or at higher doses, which indicates that this material may cause DNA damage and/or may have mutagenic effects. Consequently, its use should be restricted. Its chloroform extract (CE) obtained from the leaves demonstrated potential as anticancer and anti-HPV therapeutic for treatment of HPV infection and cervical cancer [166]. The cytotoxic activities of the EE of *K. pinnata* leaves were compared with the EE of three other species of the genus (*K. daigremontiana*, *K. milloti*, *K. nyikae*). The EE of *K. pinnata* showed the highest cytotoxicity against a lymphoma cell line, in a dose-dependent manner [135]. The anticancer mechanisms were revealed through a molecular approach [167] to support the use of *K. pinnata* as an adjuvant in cancer treatment. Gallic acid, caffeic acid, coumaric acid, quercetin, quercitrin, isorhamnetin, kaempferol, bersaldegenin, bryophyllin A, bryophyllin C, bryophynol, bryophyllol, bryophollone, stigmasterol, and campesterol were identified as bioactive compounds which participate. Some compounds were identified as bioactive, participating in the regulation of proliferation, apoptosis, cell migration, angiogenesis, metastasis, oxidative stress, and autophagy, with the potential to act as epigenetic drugs by reverting the acquired epigenetic changes associated with tumor resistance to therapy—such as the promoter methylation of suppressor genes, inhibition of DNMT1 and DNMT3b activity, and HDAC regulation—through methylation, thereby regulating the expression of genes involved in the PI3K/Akt/mTOR, Nrf2/Keap1, MEK/ERK, and Wnt/β-catenin pathways. Bryophyllin A, bryophyllin B, and bersaldegenin-3-acetate isolated from AE of *K. pinnata* are well known regarding their cytotoxic effects against A-549, HCT-8, P-388, and L-1210 tumor cells [114].

Two creams containing the AE of *K. pinnata* leaves (6%) and its major flavonoid quercetin 3-*O*-α-l-arabinopyranosyl-(1→2)-α-l-rhamnopyranoside (**45**) (0.15%) were developed and compared [67]. Both creams were topically evaluated and resulted in better re-epithelialization and dense collagen fibers. The flavonoid plays a fundamental role in wound healing but similar results that were found for both creams indicate that the use of the AE could be more profitable than the isolated compound. This extract of the AE also significantly prevented the increase of systolic and diastolic arterial pressures in salt hypertensive rats, and the concomitant administration of high-salt + the AE significantly prevented salt-induced hypertension in rats [65].

The effects of pressed juice (PJ), flavonoid-enriched fractions (FEF), bufadienolide-enriched fractions (BEF), and a flavonoid aglycone mixture (FAM) on detrusor contractility were investigated as a major target in overactive bladder disease [60]. The PJ increased the contraction force of muscle strips, the FEF had almost no effect on contractility, while the BEF and FAM led to a dose-dependent lowering of contraction force. The data indicated that several components of the PJ may contribute to the inhibitory effect on detrusor contractility, which in turn provides support for overactive bladder treatment. Other authors aimed to substantiate the use of the PJ [89] and AE in the treatment of premature labor [168] and in the uterine contractility [169]. In the first case, several fractions and compounds obtained from the PJ led to a dose-dependent decrease of oxytocin signaling (induced by an increase in free calcium concentration), but none was as strong as the PJ. However, the combination of a BEF and a FEF was as effective as the PJ, and the combination had a synergistic effect. The PJ inhibited oxytocin-driven activation, and this effect was comparable to that of the Atosiban oxytocin-receptor antagonist and tocolytic agent. In the second case, the AE showed to be as effective as beta-agonists, but significantly better tolerated. The antioxidant activity of 34 juices of species of the *Kalanchoe* genus were also compared, and the species *K. scapigera* and *K. rhombopilosa* showed the highest antioxidant activity (1981 mg/L and 1911 mg/L, respectively) [170]. A market product from PJ of *K. pinnata* was tested in prospective-observational studies in pregnancy [171], in patients with cancer and suffering from sleep problems [172], and the results suggested that these tablets can be a suitable treatment in both conditions.

The anticonvulsant activity from the ME of roots and stems of *K. pinnata* decreased with increased doses of the ME of roots, whereas the effect of the ME of stems was dose-dependent (it increased with higher doses) and this effect was preserved when the mixture of chloroform and ethyl acetate were tested. The dose of 400 mg/kg of the ME significantly improved the memory and learning of mice [62,80]. A study utilizing a larval zebrafish model was conducted to assess the potential of the AE obtained from *K. pinnata* leaves; the results indicated that the AE exhibited both anxiolytic and psychoactive effects, in a dose-dependent manner [173]. The findings of this study contribute to a deeper understanding of the underlying mechanisms responsible for these behavioral effects, thereby providing valuable insights that support the safe and effective utilization of AE in the treatment of mood disorders.

There are few studies about the isolation and characterization of bioactive molecules from *Kalanchoe* species correlated to their pharmacological potential. Detailed information regarding the studies reported in the literature can be observed in Table 7. 

## 3. Methodology

This literature review used published scientific materials collected from the PubMed^®^ and SciFinder^®^ databases without restriction regarding the year of publication and includes literature published through April of 2023. The search term used was “*Kalanchoe*”. The chemical names agree with the original references.

## 4. Conclusions and Future Perspectives

This review describes the popular uses, anatomical, and biological aspects of the *Kalanchoe* species, a plant genus widely prescribed in folk medicine and popularly known as the “miracle leaf”. Even though the *Kalanchoe* genus has 133 accepted species names, only 19 species with popular uses have been described in the literature; 16 species have received botanical and pharmacological evaluation and only 6 species have received some chemical research in relation to isolated compounds. The species are mainly used in folk medicine to treat wounds, cancer, diabetes, infections, and inflammation. However, in the pharmacological evaluation, these species were not always studied in these models. Of the four species with the highest incidence of popular medicinal use, only *K. pinnata* was tested in relation to cutaneous wounds and the re-epithelialization process, and diabetes. The others (*K. crenata*, *K. laciniata*, and *K. daigremontiana*) have not yet been studied, but are popularly reported with these uses. *Kalanchoe crenata*, for example, has only been evaluated for cytotoxicity so far, but is often recommended for wound treatment as well as for diabetes, infections, and inflammation. All parts of the plant are utilized but the juice or crude extract are most widely used. The most utilized species are *K. pinnata*, *K. crenata*, *K. laciniata*, and *K. daigremontiana*. The literature does not describe which parts of the plant or methods of preparation are popularly recommended for medicinal use in relation to *K.* × *houghtonii*, *K. flammea*, *K. gastonis-bonnieri*, and *K. integra*. Several species have structural similarities, although few of them have macroscopic or microscopic information described in the literature. Further studies are necessary to differentiate the species. One hundred and twenty-three compounds were isolated from the *Kalanchoe* genus, mainly phenols, cardiac glycosides, and triterpenes. Most of the compounds were isolated from *K. daigremontiana*, *K. pinnata*, *K. delagoensis*, and *K. ceratophylla*. Pharmacological studies have validated antioxidant, anti-inflammatory, cytotoxic, and antimicrobial activities, some of which are related to ethnopharmacological uses. Of the sixteen studied species four are not reported in the literature regarding their popular uses; however, they have been tested regarding pharmacological activities (*K. blossfeldiana*, *K. longiflora*, *K. scapigera*, and *K. rhombopilosa*). More in vivo studies should be conducted to obtain information about the bioavailability of the chemical compounds present in the extracts, and to propose active doses of these extracts that could be used in vivo to promote the expected biological activities. These experiments could also help to determine the toxicity of these doses, and the possible adverse effects that might be related to these bioactive compounds. Analytical experiments to standardize the extracts and identify possible chemical markers that could be used for quality control are also required. Finally, the authors consider that pharmacological studies dealing with yet unexplored areas should be encouraged to increase other possible medicinal uses of the extracts of these species of *Kalanchoe*.

## Figures and Tables

**Figure 1 molecules-28-05574-f001:**
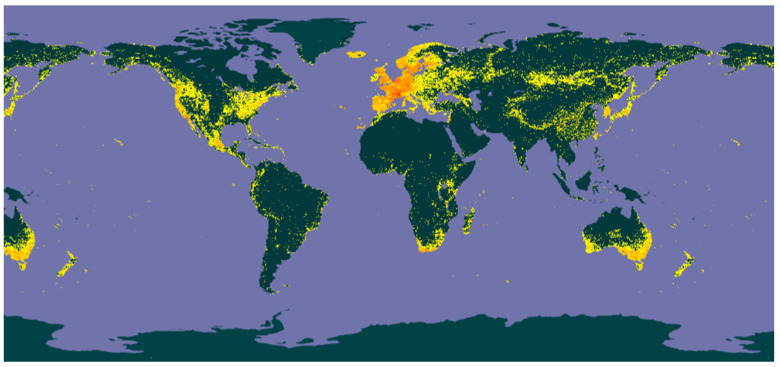
Distribution of the species of the family Crassulaceae (yellow spots).

**Figure 2 molecules-28-05574-f002:**
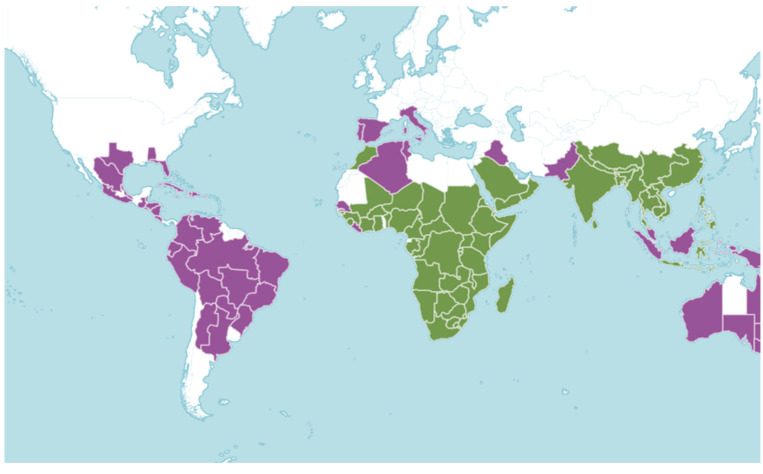
Distribution of the native (green) and introduced (purple) species of the genus *Kalanchoe*.

**Figure 3 molecules-28-05574-f003:**
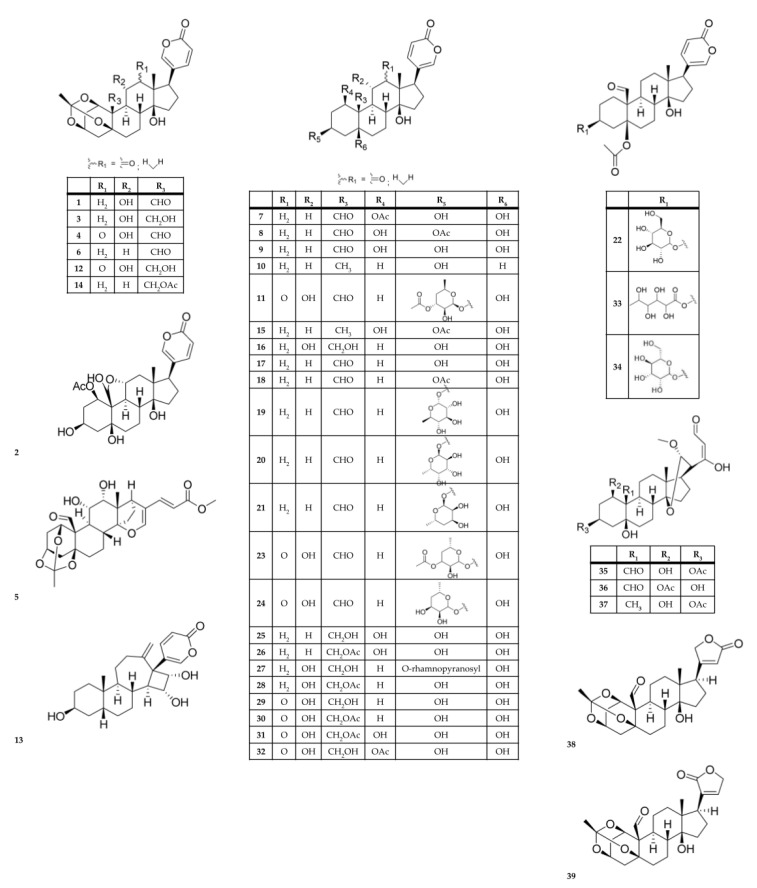
Chemical structures of cardiac glycosides from *Kalanchoe* species.

**Figure 4 molecules-28-05574-f004:**
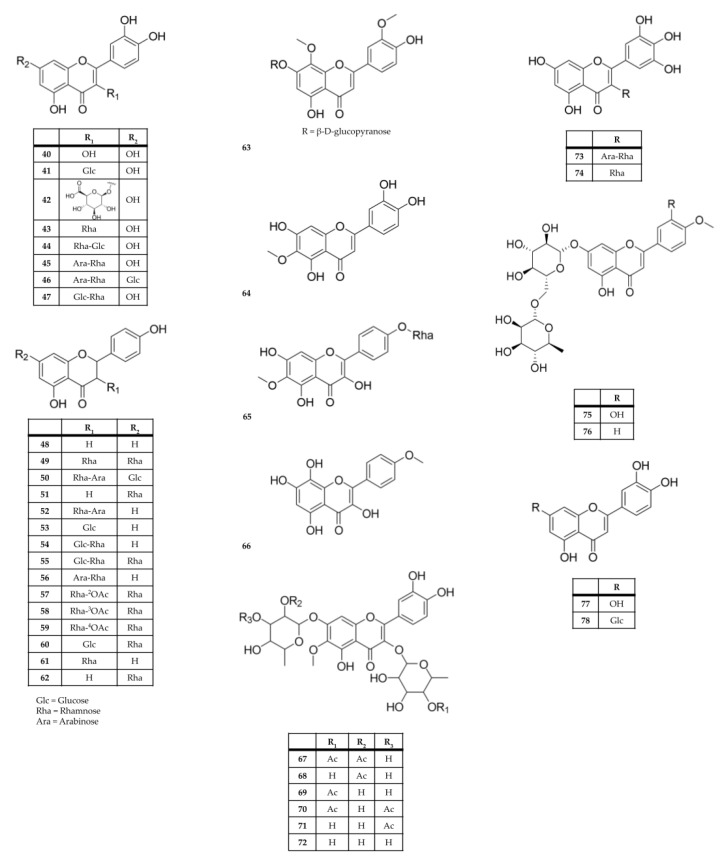
Chemical structures of flavonoids from *Kalanchoe* species.

**Figure 5 molecules-28-05574-f005:**
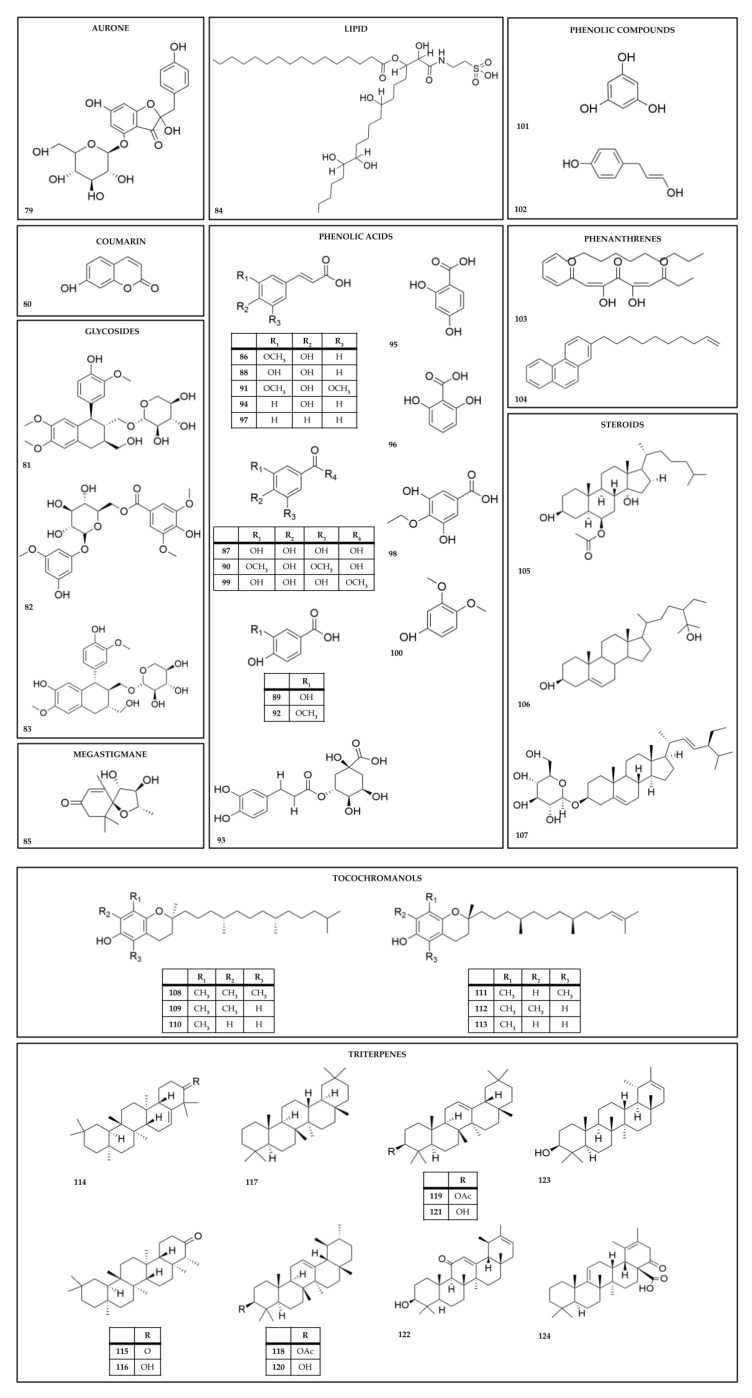
Chemical structures of other secondary metabolites from *Kalanchoe* species.

**Table 1 molecules-28-05574-t001:** Kalanchoe species, synonyms, and number of occurrences.

Scientific Name	Synonym	Occurrences
*Kalanchoe adelae* Raym.-Hamet	*Bryophyllum adelae* (Raym.-Hamet) A.Berger, *Kalanchoe floribunda* Tul.	2
*Kalanchoe aliciae* Raym.-Hamet	*Bryophyllum aliciae* (Raym.-Hamet) A.Berger, *Kalanchoe miniata* var. *tsinjoarivensis* H.Perrier, *Kalanchoe pubescens* var. *brevicalyx* Boiteau and Mannoni, *Kalanchoe pubescens* var. *grandiflora* Boiteau and Mannoni	4
*Kalanchoe alternans* (Vahl) Pers.	*Cotyledon alternans* Vahl, *Vereia alternans* (Vahl) Spreng.	56
*Kalanchoe alticola* Compton	-	5
*Kalanchoe ambolensis* Humbert	-	10
*Kalanchoe angolensis* N.E.Br.	-	1
*Kalanchoe antennifera* Desc.	-	2
*Kalanchoe arborescens* Humbert	-	43
*Kalanchoe aromatica* H.Perrier	-	27
*Kalanchoe aubrevillei* Raym.-Hamet ex Cufod.	-	12
*Kalanchoe* × *auriculata* (Raadts) V.V.Byalt	*Kalanchoe nyikae* subsp. *auriculata* Raadts	47
*Kalanchoe ballyi* Raym.-Hamet ex Cufod.	-	27
*Kalanchoe beauverdii* Raym.-Hamet	*Bryophyllum beauverdii* (Raym.-Hamet) A.Berger, *Kalanchoe beauverdii* var. *typica* Boiteau and Mannoni	152
*Kalanchoe beharensis* Drake	*Kalanchoe vantieghemii* Raym.-Hamet	386
*Kalanchoe benbothae* Gideon F.Sm. and N.R.Crouch	-	6
*Kalanchoe bentii* C.H.Wright ex Hook f.	-	37
*Kalanchoe berevoensis* Rebmann	-	-
*Kalanchoe bergeri* Raym.-Hamet and H.Perrier	*Bryophyllum bergeri* (Raym.-Hamet and H.Perrier) Govaerts, *Kalanchoe bergeri* var. *typica* Boiteau and Mannoni, *Kalanchoe bergeri* var. *glabra* Boiteau and Mannoni	27
*Kalanchoe bhidei* T.Cooke	-	16
*Kalanchoe bipartita* Chiov.	-	14
*Kalanchoe blossfeldiana* Poelln.	*Kalanchoe coccinea* (H.Perrier) Boiteau, *Kalanchoe coccinea* var. *blossfeldiana* (Poelln.) Boiteau, *Kalanchoe globulifera* var. *coccinea* H.Perrier	493
*Kalanchoe bogneri* Rauh	*Bryophyllum bogneri* (Rauh) V.V.Byalt	10
*Kalanchoe boisii* Raym.-Hamet and H.Perrier	-	2
*Kalanchoe boranae* Raadts	-	10
*Kalanchoe bouvetii* Raym.-Hamet and H.Perrier	*Bryophyllum bouvetii* (Raym.-Hamet and H.Perrier) A.Berger	16
*Kalanchoe bouvetii* Raym.-Hamet and H.Perrier	*Kalanchoe baumii* Engl. and Gilg, *Kalanchoe multiflora* Schinz, *Kalanchoe pruinosa* Dinter, *Kalanchoe pyramidalis* Schönland	16
*Kalanchoe bracteata* Scott Elliot	*Kalanchoe bracteata* var. *aurantiaca* Rauh and Hebding, *Kalanchoe bracteata* var. *glabra* Rauh and Hebding, *Kalanchoe bracteata* subsp. *glabra* Rauh and Hebding, *Kalanchoe bracteata* var. *longisepala* Boiteau ex L.Allorge, *Kalanchoe bracteata* var. *pubescens* Rauh and Hebding, *Kalanchoe bracteata* var. *virescens* Desc., *Kalanchoe nadyae* Raym.-Hamet	88
*Kalanchoe brevicalyx* (Raym.-Hamet and H.Perrier) Gideon F.Sm. and Figueiredo	*Kalanchoe pinnata* var. *brevicalyx* Raym.-Hamet and H.Perrier	1
*Kalanchoe briquetii* Raym.-Hamet	-	2
*Kalanchoe campanulata* (Baker) Baill.	*Bryophyllum campanulatum* (Baker) V.V.Byalt, Udalova and I.M.Vassiljeva, *Kitchingia campanulata* Baker, *Kalanchoe amplexicaulis* (Baker) Baill., *Kalanchoe campanulata* subsp. *orthostyla* Boiteau and Mannoni, *Kalanchoe panduriformis* (Baker) Baill., *Kalanchoe parviflora* (Baker) Baill., *Kitchingia amplexicaulis* Baker, *Kitchingia panduriformis* Baker, *Kitchingia parviflora* Baker	80
*Kalanchoe ceratophylla* Haw.		111
*Kalanchoe chapototii* Raym.-Hamet and H.Perrier		2
*Kalanchoe cherukondensis* Subba Rao and Kumari	*Vereia ceratophylla* (Haw.) D.Dietr.	-
*Kalanchoe chevalieri* Gagnep.	*Kalanchoe integra* var. *chevalieri* (Gagnep.) H.H.Pham	9
*Kalanchoe citrina* Schweinf.	*Kalanchoe citrina* var. *ballyi* Raym.-Hamet ex Wickens, *Kalanchoe citrina* var. *erythreae* Schweinf.	102
*Kalanchoe costantinii* Raym.-Hamet	*Bryophyllum costantinii* (Raym.-Hamet) A.Berger	1
*Kalanchoe craibii* Raym.-Hamet	-	1
*Kalanchoe crenata* (Andrews) Haw.	*Cotyledon crenata* (Andrews) Vent., *Cotyledon verea* Jacq., *Kalanchoe afzeliana* Britten, *Kalanchoe crenata* var. *verea* Cufod., *Kalanchoe integra* var. *crenata* (Andrews) Cufod., *Kalanchoe integra* var. *varea* Cufod., *Kalanchoe verea* Pers., *Vereia crenata* Andrews	1320
*Kalanchoe crouchii* Gideon F.Sm. and Figueiredo	-	3
*Kalanchoe crundallii* I.Verd.	-	6
*Kalanchoe curvula* Desc.	*Bryophyllum curvulum* (Desc.) V.V.Byalt	16
*Kalanchoe cymbifolia* Desc.	*Bryophyllum cymbifolium* (Desc.) V.V.Byalt	-
*Kalanchoe daigremontiana* Raym.-Hamet and H.Perrier	*Bryophyllum daigremontianum* (Raym.-Hamet and H.Perrier) A.Berger	768
*Kalanchoe darainensis* D.-P.Klein and Callm.	-	22
*Kalanchoe decumbens* Compton	-	-
*Kalanchoe deficiens* (Forssk.) Asch. and Schweinf.	*Cotyledon deficiens* Forssk., *Kalanchoe glaucescens* var. *deficiens* (Asch. and Schweinf.) Senni	342
*Kalanchoe delagoensis* Eckl. and Zeyh.	*Bryophyllum delagoense* (Eckl. and Zeyh.) Druce, *Bryophyllum tubiflorum* Harv., *Kalanchoe tubiflora* (Harv.) Raym.-Hamet, *Bryophyllum verticillatum* (Scott Elliot) A.Berger, *Geaya purpurea* Costantin and Poiss., *Kalanchoe verticillata* Scott Elliot	5341
*Kalanchoe densiflora* Rolfe	-	687
*Kalanchoe* × *descoingsii* Shtein, Gideon F.Sm. and J.Ikeda	-	-
*Kalanchoe dinklagei* Rauh	*Kalanchoe brevisepala* (Humbert) L.Allorge, *Kalanchoe millotii* var. *brevisepala* Humbert	14
*Kalanchoe dyeri* N.E.Br.	-	21
*Kalanchoe elizae* A.Berger	*Cotyledon elizae* (A.Berger) Raym.-Hamet, *Cotyledon insignis* N.E.Br., *Kalanchoe insignis* (N.E.Br.) N.E.Br., *Kalanchoe laurensii* Raym.-Hamet	80
*Kalanchoe eriophylla* Hils. and Bojer ex Tul.	*Cotyledon pannosa* Baker	43
*Kalanchoe* × *estrelae* Gideon F.Sm.	-	-
*Kalanchoe fadeniorum* Raadts	-	8
*Kalanchoe farinacea* Balf.f.	-	55
*Kalanchoe faustii* Font Quer	*Kalanchoe laciniata* subsp. *faustii* (Font Quer) Maire	33
*Kalanchoe fedtschenkoi* Raym.-Hamet and H.Perrier	*Bryophyllum fedtschenkoi* (Raym.-Hamet and H.Perrier) Lauz.-March., *Kalanchoe fedtschenkoi* var. *isalensis* Boiteau and Mannoni	514
*Kalanchoe fernandesii* Raym.-Hamet	-	4
*Kalanchoe* × *flaurantia* Desc.	-	-
*Kalanchoe gastonis-bonnieri* Raym.-Hamet and H.Perrier	*Bryophyllum gastonis-bonnieri* (Raym.-Hamet and H.Perrier) Lauz.-March., *Kalanchoe adolphi-engleri* Raym.-Hamet, *Kalanchoe gastonis-bonnieri* var. *ankaizinensis* Boiteau ex L.Allorge	173
*Kalanchoe germanae* Raym.-Hamet ex Raadts	-	16
*Kalanchoe gideonsmithii* N.R.Crouch and Figueiredo	-	1
*Kalanchoe glaucescens* Britten	*Kalanchoe beniensis* De Wild., *Kalanchoe elliptica* Raadts, *Kalanchoe flammea* Stapf, *Kalanchoe holstii* Engl., *Kalanchoe magnidens* N.E.Br., *Kalanchoe marinellii* Pamp., *Kalanchoe ndorensis* Schweinf. ex Engl.	379
*Kalanchoe globulifera* H.Perrier	-	15
*Kalanchoe gracilipes* (Baker) Baill.	*Bryophyllum gracilipes* (Baker) Eggli, *Kitchingia gracilipes* Baker	89
*Kalanchoe grandidieri* Baill.	-	64
*Kalanchoe grandiflora* Wight and Arn.	*Vereia grandiflora* (Wight and Arn.) D.Dietr.	77
*Kalanchoe guignardii* Raym.-Hamet and H.Perrier	*Kalanchoe beauverdii* var. *guignardii* (Raym.-Hamet and H.Perrier) Boiteau and Mannoni	1
*Kalanchoe* × *gunniae* Gideon F.Sm. and Figueiredo	-	-
*Kalanchoe hametiorum* Raym.-Hamet	-	4
*Kalanchoe hauseri* Werderm.	-	-
*Kalanchoe hildebrandtii* Baill.	*Kalanchoe gomphophylla* Baker, *Kalanchoe hildebrandtii* var. *glabra* Rauh and Hebding	95
*Kalanchoe hirta* Harv.	-	13
*Kalanchoe* × *houghtonii* D.B.Ward	*Bryophyllum* × *houghtonii* (D.B.Ward) P.I.Forst.	1650
*Kalanchoe humifica* Desc.	*Bryophyllum humificum* (Desc.) V.V.Byalt	1
*Kalanchoe humilis* Britten	-	30
*Kalanchoe hypseloleuce* Friis and M.G.Gilbert	-	1
*Kalanchoe inaurata* Desc.	*Bryophyllum inauratum* (Desc.) V.V.Byalt	-
*Kalanchoe integra* (Medik.) Kuntze	*Cotyledon integra* Medik., *Bryophyllum serratum* Blanco, *Cotyledon acutiflora* (Haw.) W.T.Aiton, *Cotyledon hybrida* Dum.Cours., *Cotyledon spathulata* (DC.) Poir., *Echeveria spathulata* (DC.) W.Bull ex É.Morren, *Kalanchoe acutiflora* (Andrews) Haw., *Kalanchoe annamica* Gagnep., *Kalanchoe corymbosa* Wall., *Kalanchoe dixoniana* Raym.-Hamet, *Kalanchoe garambiensis* Kudô, *Kalanchoe hybrida* Desf. ex Steud., *Kalanchoe integra* var. *annamica* (Gagnep.) H.H.Pham, *Kalanchoe nudicaulis* Buch.-Ham. ex C.B.Clarke, *Kalanchoe schumacheri* Koord., *Kalanchoe spathulata* DC., *Kalanchoe spathulata* var. *annamica* (Gagnep.) H.Ohba, *Kalanchoe spathulata* var. *baguioensis* H.Ohba, *Kalanchoe spathulata* var. *ciliata*, *Kalanchoe spathulata* var. *dixoniana* (Raym.-Hamet) H.Ohba, *Kalanchoe spathulata* var. *garambiensis* (Kudô) H.Ohba, *Kalanchoe spathulata* var. *schumacheri* (Koord.) H.Ohba, *Kalanchoe spathulata* var. *simlensis* H.Ohba, *Kalanchoe spathulata* var. *staintonii* H.Ohba, *Kalanchoe subamplectens* Wall., *Kalanchoe varians* Haw., *Kalanchoe yunnanensis* Gagnep., *Vereia acutiflora* Andrews, *Vereia spathulata* (DC.) D.Dietr.	289
*Kalanchoe integrifolia* Baker	*Kalanchoe bitteri* Raym.-Hamet and H.Perrier, *Kalanchoe heckelii* Raym.-Hamet and H.Perrier, *Kalanchoe integrifolia* var. *bitteri* Raym.-Hamet and H.Perrier, *Kalanchoe integrifolia* var. *flava* Boiteau	85
*Kalanchoe jongmansii* Raym.-Hamet and H.Perrier	*Bryophyllum jongmansii* (Raym.-Hamet and H.Perrier) Govaerts, *Kalanchoe jongmansii* subsp. *ivohibensis* Humbert	54
*Kalanchoe klopperae* Gideon F.Sm. and Figueiredo	-	-
*Kalanchoe laciniata* (L.) DC.	*Cotyledon laciniata* L., *Vereia laciniata* (L.) Willd., *Kalanchoe angustifolia* A.Rich., *Kalanchoe biternata* Wight ex Wall., *Kalanchoe carnea* N.E.Br., *Kalanchoe gloveri* Cufod., *Kalanchoe lentiginosa* Cufod., *Kalanchoe petitiaesii* Rich. ex Jacques, *Kalanchoe rohlfsii* Engl., *Kalanchoe rosea* A.Chev., *Kalanchoe schweinfurthii* Penz., *Kalanchoe teretifolia* Haw.	430
*Kalanchoe laetivirens* Desc.	*Bryophyllum laetivirens* (Desc.) V.V.Byalt	223
*Kalanchoe lanceolata* (Forssk.) Pers.	*Cotyledon lanceolata* Forssk., *Vereia lanceolata* (Forssk.) Spreng., *Cotyledon amplexicaulis* B.Heyne ex C.B.Clarke, *Cotyledon corymbosa* Rottler ex Wight and Arn., *Cotyledon heterophylla* Roxb., *Cotyledon hirsuta* B.Heyne ex C.B.Clarke, *Cotyledon paniculata* Rottler ex Wight and Arn., *Kalanchoe amplexicaulis* B.Heyne, *Kalanchoe brachycalyx* A.Rich., *Kalanchoe crenata* var. *collina* Engl., *Kalanchoe ellacombei* N.E.Br., *Kalanchoe floribunda* Wight and Arn., *Kalanchoe floribunda* var. *glabra* C.B.Clarke, *Kalanchoe glandulosa* Hochst. ex A.Rich., *Kalanchoe glandulosa* var. *benguellensis* Engl., *Kalanchoe glandulosa* var. *rhodesica* Baker f., *Kalanchoe glandulosa* var. *tomentosa* Keissl., *Kalanchoe goetzei* Engl., *Kalanchoe gregaria* Dinter, *Kalanchoe heterophylla* (Roxb.) Wight and Arn., *Kalanchoe heterophylla* (Roxb.) Prain, *Kalanchoe homblei* De Wild., *Kalanchoe homblei* f. *reducta* De Wild., *Kalanchoe junodii* Schinz, *Kalanchoe laciniata* var. *brachycalyx* (A.Rich.) Chiov., *Kalanchoe lanceolata* var. *glabra* (C.B.Clarke) S.R.Sriniv., *Kalanchoe lanceolata* var. *glandulosa* (Hochst. ex A.Rich.) Cufod., *Kalanchoe modesta* Kotschy and Peyr., *Kalanchoe pentheri* Schltr., *Kalanchoe pilosa* Baker, *Kalanchoe platysepala* Welw. ex Britten, *Kalanchoe pubescens* R.Br. ex Britten, *Kalanchoe ritchieana* Dalzell, *Kalanchoe spathulata* Wall., *Kalanchoe wightianum* Wall., *Meristostylus macrocalyx* Klotzsch, *Vereia floribunda* (Wight and Arn.) D.Dietr., *Vereia heterophylla* (Wight and Arn.) D.Dietr.	893
*Kalanchoe lateritia* Engl.	-	284
*Kalanchoe latisepala* N.E.Br.	-	31
*Kalanchoe laxiflora* Baker	*Bryophyllum laxiflorum* (Baker) Govaerts, *Bryophyllum crenatum* Baker, *Kalanchoe crenata* (Baker) Raym.-Hamet, *Kalanchoe laxiflora* subsp. *stipitata* Boiteau and Mannoni, *Kalanchoe laxiflora* subsp. *subpeltata* Boiteau and Mannoni, *Kalanchoe laxiflora* subsp. *violacea* Boiteau and Mannoni, *Kalanchoe tieghemii* Raym.-Hamet	469
*Kalanchoe leblanciae* Raym.-Hamet	-	17
*Kalanchoe lindmanii* Raym.-Hamet	*Kalanchoe gossweileri* Croizat, *Kalanchoe humbertii* Guillaumin, *Kalanchoe pearsonii* N.E.Br.	16
*Kalanchoe linearifolia* Drake	*Kalanchoe bonnieri* Raym.-Hamet	118
*Kalanchoe lobata* R.Fern.	-	6
*Kalanchoe* × *lokarana* Desc.	*Bryophyllum* × *lokarana* (Desc.) V.V.Byalt	2
*Kalanchoe longiflora* Schltr.	-	35
*Kalanchoe longifolia* E.T.Geddes	-	2
*Kalanchoe lubangensis* R.Fern.	-	1
*Kalanchoe luciae* Raym.-Hamet	*Kalanchoe albiflora* H.M.L.Forbes	60
*Kalanchoe macrochlamys* H.Perrier	*Bryophyllum macrochlamys* (H.Perrier) A.Berger	12
*Kalanchoe mandrarensis* Humbert	-	7
*Kalanchoe manginii* Raym.-Hamet and H.Perrier	*Bryophyllum manginii* (Raym.-Hamet and H.Perrier) Nothdurft	67
*Kalanchoe marmorata* Baker	*Kalanchoe grandiflora* A.Rich., *Kalanchoe macrantha* Baker ex Maire, *Kalanchoe macrantha* var. *marmorata* (Baker) Maire, *Kalanchoe macrantha* var. *richardiana* Maire	264
*Kalanchoe marnieriana* H.Jacobsen ex L.Allorge	*Bryophyllum marnierianum* (H.Jacobsen ex L.Allorge) Govaerts, *Kalanchoe humbertii* Mannoni and Boiteau	46
*Kalanchoe maromokotrensis* Desc. and Rebmann	-	5
*Kalanchoe migiurtinorum* Cufod.	-	7
*Kalanchoe millotii* Raym.-Hamet and H.Perrier	-	82
*Kalanchoe miniata* Hils. and Bojer ex Tul.	*Bryophyllum miniatum* (Hils. and Bojer ex Tul.) A.Berger, *Kalanchoe miniata* var. *typica* H.Perrier, *Kitchingia miniata* (Hils. and Bojer ex Tul.) Baker	252
*Kalanchoe mitejea* Leblanc and Raym.-Hamet	-	29
*Kalanchoe montana* Compton	*Kalanchoe luciae* subsp. *montana* (Compton) Toelken	2
*Kalanchoe mortagei* Raym.-Hamet and H.Perrier	*Bryophyllum mortagei* (Raym.-Hamet and H.Perrier) Wickens, *Kalanchoe poincarei* var. *mortagei* (Raym.-Hamet and H.Perrier) Boiteau	42
*Kalanchoe ndotoensis* L.E.Newton	-	1
*Kalanchoe neglecta* Toelken	*Kalanchoe rotundifolia* f. *peltata* R.Fern.	7
*Kalanchoe nyikae* Engl.	*Kalanchoe hemsleyana* Cufod.	53
*Kalanchoe obtusa* Engl.	-	39
*Kalanchoe olivacea* Dalzell	-	10
*Kalanchoe orgyalis* Baker	*Kalanchoe antanosiana* Drake	162
*Kalanchoe paniculata* Harv.	*Sedum harveyanum* Kuntze, *Kalanchoe oblongifolia* Harv.	193
*Kalanchoe pareikiana* Desc. and Lavranos	-	2
*Kalanchoe peltata* (Baker) Baill.	*Bryophyllum peltatum* (Baker) V.V.Byalt, Udalova and I.M.Vassiljeva, *Kitchingia peltata* Baker	152
*Kalanchoe peltigera* Desc.	*Bryophyllum peltigerum* (Desc.) V.V.Byalt	5
*Kalanchoe perrieri* Shtein, Gideon F.Sm. and D.-P.Klein	-	-
*Kalanchoe peteri* Werderm.	-	35
*Kalanchoe petitiana* A.Rich.	-	95
*Kalanchoe pinnata* (Lam.) Pers.	*Bryophyllum pinnatum* (Lam.) Oken, *Cotyledon pinnata* Lam., *Crassula pinnata* (Lam.) L.f., *Kalanchoe pinnata* var. *genuina* Raym.-Hamet, *Vereia pinnata* (Lam.) Spreng., *Baumgartenia sobolifera* Tratt., *Bryophyllum calcicola* (H.Perrier) V.V.Byalt, *Bryophyllum calycinum* Salisb., *Bryophyllum germinans* Blanco, *Bryophyllum pinnatum simplicifolium* Kuntze, *Cotyledon calycina* (Salisb.) B.Heyne, *Cotyledon calyculata* Sol. ex Sims, *Cotyledon rhizophylla* Roxb., *Crassuvia floripendia* Comm. ex Lam., *Kalanchoe calcicola* (H.Perrier) Boiteau, *Kalanchoe floripendula* Steud, *Kalanchoe pinnata* var. *calcicola* H.Perrier, *Kalanchoe pinnata* var. *floripendula* Pers.	7288
*Kalanchoe* × *poincarei* Raym.-Hamet and H.Perrier	*Bryophyllum poincarei* (Raym.-Hamet and H.Perrier) Govaerts	10
*Kalanchoe porphyrocalyx* (Baker) Baill.	*Bryophyllum porphyrocalyx* (Baker) A.Berger, *Kalanchoe porphyrocalyx* var. *typica* Boiteau and Mannoni, *Kitchingia porphyrocalyx* Baker	187
*Kalanchoe prasina* N.E.Br.	*Kalanchoe figuereidoi* Croizat	-
*Kalanchoe prittwitzii* Engl.	*Kalanchoe dielsii* Raym.-Hamet, *Kalanchoe lugardii* Bullock, *Kalanchoe robynsiana* Raym.-Hamet, *Kalanchoe secunda* Werderm.	136
*Kalanchoe prolifera* (Bowie ex Hook.) Raym.-Hamet	*Bryophyllum proliferum* Bowie ex Hook., *Bryophyllum cochleatum* Lem., *Kalanchoe cochleatum* (Lem.) B.D.Jacks.	180
*Kalanchoe pseudocampanulata* Mannoni and Boiteau	*Bryophyllum pseudocampanulatum* (Mannoni and Boiteau) Govaerts, *Kalanchoe miniata* var. *decaryana* H.Perrier	5
*Kalanchoe pubescens* Baker	*Bryophyllum pubescens* (Baker) Govaerts, *Kalanchoe pubescens* var. *typica* Boiteau and Mannoni	162
*Kalanchoe pumila* Baker	*Kalanchoe brevicaulis* Baker, *Kalanchoe multiceps* Baill., *Kalanchoe pumila* f. *venustior* Boiteau	71
*Kalanchoe quadrangularis* Desc.	-	3
*Kalanchoe quartiniana* A.Rich.	-	23
*Kalanchoe rebmannii* Desc.	-	1
*Kalanchoe* × *rechingeri* Raym.-Hamet ex Rauh and Hebding	*Bryophyllum* × *rechingeri* (Raym.-Hamet ex Rauh and Hebding) V.V.Byalt	2
*Kalanchoe rhombopilosa* Mannoni and Boiteau	*Kalanchoe rhombopilosa* var. *argentea* Rauh, *Kalanchoe rhombopilosa* var. *viridifolia* Rauh	30
*Kalanchoe* × *richaudii* Desc.	-	2
*Kalanchoe robusta* Balf.f.	*Kalanchoe abrupta* Balf.f.	7
*Kalanchoe rolandi-bonapartei* Raym.-Hamet and H.Perrier	*Bryophyllum rolandi-bonapartei* (Raym.-Hamet and H.Perrier) Govaerts, *Bryophyllum tsaratananense* (H.Perrier) A.Berger, *Kalanchoe tsaratananensis* H.Perrier	16
*Kalanchoe rosea* C.B.Clarke	-	-
*Kalanchoe rosei* Raym.-Hamet and H.Perrier	*Bryophyllum rosei* (Raym.-Hamet and H.Perrier) A.Berger, *Kalanchoe bouvieri* Raym.-Hamet and H.Perrier	74
*Kalanchoe rotundifolia* (Haw.) Haw.	*Crassula rotundifolia* Haw., *Sedum subrotundifolium* (Haw.) Kuntze, *Vereia rotundifolia* (Haw.) D.Dietr., *Kalanchoe guillauminii* Raym.-Hamet, *Kalanchoe integerrima* Lange, *Kalanchoe luebbertiana* Engl., *Kalanchoe rotundifolia* var. *guillauminii* (Raym.-Hamet) Raym.-Hamet, *Kalanchoe rotundifolia* f. *tripartita* R.Fern., *Kalanchoe seilleana* Raym.-Hamet, *Kalanchoe stearnii* Raym.-Hamet, *Meristostylus brachycalyx* Klotzsch	623
*Kalanchoe rubella* (Baker) Raym.-Hamet	*Bryophyllum rubellum* Baker	23
*Kalanchoe salazarii* Raym.-Hamet	-	2
*Kalanchoe sanctula* Desc.	*Bryophyllum sanctulum* (Desc.) V.V.Byalt	2
*Kalanchoe scandens* H.Perrier	*Bryophyllum scandens* (H.Perrier) A.Berger, *Kalanchoe beauverdii* var. *parviflora* Boiteau and Mannoni	7
*Kalanchoe scapigera* Welw. ex Britten	-	15
*Kalanchoe schimperiana* A.Rich.	*Cotyledon deficiens* Hochst. and Steud. ex A.Rich.	78
*Kalanchoe schizophylla* (Baker) Baill.	*Bryophyllum schizophyllum* (Baker) A.Berger, *Kitchingia schizophylla* Baker	48
*Kalanchoe schliebenii* Werderm.	-	3
*Kalanchoe serrata* Mannoni and Boiteau	*Bryophyllum lauzac-marchaliae* V.V.Byalt, *Bryophyllum serratum* (Mannoni and Boiteau) Lauz.-March.	36
*Kalanchoe sexangularis* N.E.Br.	-	130
*Kalanchoe stenosiphon* Britten	-	9
*Kalanchoe streptantha* Baker	*Bryophyllum streptanthum* (Baker) A.Berger	28
*Kalanchoe suarezensis* H.Perrier	*Bryophyllum suarezense* (H.Perrier) A.Berger, *Kalanchoe poincarei* var. *suarezensis* (H.Perrier) L.Allorge	20
*Kalanchoe subrosulata* Thulin	-	4
*Kalanchoe synsepala* Baker	*Kalanchoe brachycalyx* Baker, *Kalanchoe gentyi* Raym.-Hamet and H.Perrier, *Kalanchoe trichantha* Baker	214
*Kalanchoe tachingshuii* S.S.Ying	-	-
*Kalanchoe tashiroi* Yamam.	-	4
*Kalanchoe teixeirae* Raym.-Hamet ex R.Fern.	-	3
*Kalanchoe tenuiflora* Desc.	-	3
*Kalanchoe tetramera* E.T.Geddes	-	2
*Kalanchoe tetraphylla* H.Perrier	-	30
*Kalanchoe thyrsiflora* Harv.	*Kalanchoe alternans* Eckl. and Zeyh. ex Harv.	251
*Kalanchoe tomentosa* Baker	*Bryophyllum triangulare* Blanco	179
*Kalanchoe torrejacqii* Shtein and Gideon F.Sm.	-	3
*Kalanchoe tuberosa* H.Perrier	-	11
*Kalanchoe uniflora* (Stapf) Raym.-Hamet	*Bryophyllum uniflorum* (Stapf) A.Berger, *Kitchingia uniflora* Stapf, *Bryophyllum ambrense* (H.Perrier) A.Berger, *Kalanchoe ambrensis* H.Perrier, *Kalanchoe uniflora* var. *brachycalyx* Boiteau and Mannoni	97
*Kalanchoe usambarensis* Engl. and Raym.-Hamet	-	16
*Kalanchoe variifolia* (Guillaumin and Humbert) Shtein, D.-P.Klein and Gideon F.Sm.	*Kalanchoe rosei* var. *variifolia* (Guillaumin and Humbert) J.M.H.Shaw, *Kalanchoe rosei* subsp. *variifolia* Guillaumin and Humbert	13
*Kalanchoe velutina* Welw. ex Britten	-	56
*Kalanchoe viguieri* Raym.-Hamet and H.Perrier	*Kalanchoe viguieri* var. *latisepala* Raym.-Hamet and H.Perrier	68
*Kalanchoe waldheimii* Raym.-Hamet and H.Perrier	*Bryophyllum waldheimii* (Raym.-Hamet and H.Perrier) Lauz.-March.	50
*Kalanchoe waterbergensis* van Jaarsv.	-	3
*Kalanchoe welwitschii* Britten	-	17
*Kalanchoe wildii* Raym.-Hamet ex R.Fern.	*Kalanchoe aleuroides* Stearn	2
*Kalanchoe winteri* Gideon F.Sm., N.R.Crouch and Mich.Walters	-	3
*Kalanchoe yemensis* (Deflers) Schweinf.	*Kalanchoe brachycalyx* var. *yemensis* Deflers	17

**Table 2 molecules-28-05574-t002:** Traditional uses of *Kalanchoe* species.

Species	Traditional Uses	Form of Use and Plant Part	References
*K. ceratophylla*	To treat injuries, pain, fever, and inflammation.	Internal or external administration of crude extracts or plant juice.	[14,15,16,17]
*K. crenata*	Antidiabetic, anti-inflammatory, antimicrobial, vermifuge, and anti-infective agent; to treat wounds, abscesses, abdominal pain, asthma, headache, convulsion, smallpox, peptic ulcer, upper respiratory tract infections, coughs, otitis, palpitations, cancer (or disease states with symptoms related to cancer), diabetes, swollen areas for muscle sprain and myalgia; and to heal umbilical cord wounds in newborns.	Internal administration of crude extracts, plant juice, leaves juice, or chew the leaves; external administration of crude extracts or plant juice and from macerating the leaves into a cream. Use of roots.	[6,14,15,16,18,19,20,21,22,23,24,25,26,27]
*K. daigremontiana*	Anticancer, anti-inflammatory, antimicrobial, antiseptic, carminative and cardioactive agent; to treat skin injuries and wounds; to staunch bleeding; to treat infections, rheumatism, earache, burns, arthritis, gastric and menstrual disorders, cough, fever, cardiovascular dysfunction, diabetes, psychic agitation, restlessness and anxiety, some cancers; a chemo preventive.	Internal or external administration of crude extracts or plant juice and use of roots.	[9,10,11,12,15,21,28,29,30]
*K. delagoensis*	To treat wounds, epilepsy, neoplastic diseases, fever, abscesses, bruises, pneumonia, coughs, stomachache, and as a vermifuge.	Internal or external administration of crude extracts or plant juice and use of roots.	[14,15,21,31,32,33,34]
*K. densiflora*	To treat wounds and skin disorders, rheumatism, hemorrhoids, eye problems, joint and muscle pains, stomach and liver problems, umbilical cord, cardiac disorders, edema, poisonous, abortifacient.	Internal or external administration of crude extracts or plant juice.	[15,24,27,35]
*K. flammea*	To treat fever, wounds, inflammation, and cancer.		[36]
*K. fedtschenkoi*	Analgesic, cytotoxic, and antimicrobial treatments.	Internal or external administration. Use of leaves and roots.	[21,37]
*K. gastonis-bonnieri*	To treat genital-urinary and vaginal infections and as a vaginal contraceptive.		[38]
*K. germanae*	After removal of ganglions the leaves are used to treat the affected area.	Internal or external administration of crude extracts or plant juice.	[15]
*K. glaucescens*	To treat coughs and rheumatism.	Internal or external administration of crude extracts or plant juice.	[15,24]
*K. integra*	Antihypertensive.		[39]
*K. laciniata*	As an anti-inflammatory, astringent, and antiseptic; to treat wounds, inflammation, headache, diabetes, heart discomfort, gastric disorders, lithiasis, diarrhea, fever, cough, snakebites, erysipelas, boils, and human prostate cancer.	Internal administration of crude extracts, plant juice, leaves juice or chew the leaves; external administration of crude extracts or plant juice and from macerating the leaves into a cream.	[6,11,15,40,41,42,43,44,45,46,47,48,49]
*K. lanceolata*	To treat dysentery, rheumatism, hemorrhoids, splenomegaly, hepatomegaly, and pains.	Internal or external administration of crude extracts or plant juice.	[15,24,27,35]
*K. marmorata*	To treat wounds, boils, bruises, periodontal disease, cracked lips, arthritis, gastric ulcers, ear diseases, eye infections, dysentery, fever, common cold, coughs, cholera, urinary diseases, stiff muscles, liver problems, and headaches.	Internal or external administration of crude extracts or plant juice.	[15,24,50,51,52]
*K. mortagei*	As an antimicrobial; to treat digestive disorders, parasites, and neoplastic diseases orally; and as a local remedy for cancer.	Internal or external administration. Use of leaves and roots.	[21,37]
*K. obtusa*	Children’s diseases and as pesticide.	Use the whole plant.	[24]
*K. petitiana*	To treat epilepsy, trachoma, allergies, intestinal parasites, gonorrhea, bone setting after fractures, wound healing, breast tumors, skin cancer, swelling of gland/lymph adenitis, toothache, dysentery, liver problems, stomachache, tonsillitis, gastritis, peptic ulcer disease, and foot problems (fungal nails, corns, and calluses, athlete’s foot, plantar warts).	Internal or external administration of crude extracts or plant juice.	[15,53,54,55,56]
*K. pinnata*	Antipyretic, antibacterial, antiseptic, antimalaria, anti-inflammatory, and antipsychotic agent. To treat the following: wounds, burns; cardiovascular dysfunctions; cancer; rheumatoid arthritis; digestive, menstrual and psychiatric disorders; hypertension; skin, respiratory and genitourinary infections; kidney, liver and urinary disorders; ear, head and toothache; insect, snake and scorpion bites; muscle bruises; cholera; leishmania; leprosy; lithiasis; viruses; restlessness; biostimulator during skin transplantation; to prevent premature labor and help women recover after childbirth; diabetes; cold, whooping cough, bone fracture, Chikungunya virus, and against COVID-19 symptoms.	Internal administration of crude extracts, whole plant, or leaves juice, chew the leaves or leaves infusion; external administration of crude extracts or plant juice and from macerating the leaves into a cream. Use of roots.	[6,8,9,14,15,21,41,42,43,44,57,58,59,60,61,62,63,64,65,66,67,68,69,70,71,72,73,74,75,76,77,78,79,80,81,82,83,84,85,86,87,88,89,90,91,92,93]
*K. prittwitzii*	Stiff joints and rheumatism.	Use of leaves.	[24]
*K. serrata*	To treat pain, inflammation, fever, and viruses.	Use of leaves and roots.	[21]
*K.* x *houghtonii*	To treat infections, rheumatism, coughs, fever, and inflammation.		[30]

**Table 7 molecules-28-05574-t007:** Biological activities of compounds isolated from *Kalanchoe* species.

Species Plant Part	Compound Tested	Pharmacological Activity	Results	Mechanisms of Action	References
*K. delagoensis* Whole plant	quercetin (**40**) (6S,7R,8R,9S)-6-oxaspiro-7,8-dihydroxymegastigman-4-en-3-one (tubiflorone) (**85**) syringic acid (**90**) 4-O-ethylgallic acid (**98**) 3,4-dimethoxyphenol (**100**) phloroglucinol (**101**) 3,4-dihydroxyallylbenzene (**102**)	Anti-inflammatory Lipopolysaccharide (LPS)-induced nitric oxide (NO) production in RAW264.7 cells	The compounds demonstrated dose-dependent relationships for LPS-induced NO production. The MTT assay showed high cell viability in the presence of LPS in the culture medium at various concentrations. The results showed that quercetin (**40**), and 3,4-dihydroxyallylbenzene (**102**) possessed NO inhibitory activities, whereas (6S,7R,8R,9S)-6-oxaspiro-7,8-dihydroxymegastigman-4-en-3-one (**85**), syringic acid, and 3,4-dimethoxyphenol (**100**) exhibited weak activities.	Not reported	[32]
	kalantubolide A (**38**) kalantubolide B (**39**) kalantuboside A (**23**) kalantuboside B (**24**) bryotoxin C (**1**) bersaldegenin-1,3,5-orthoacetate (**6**) bersaldegenin-1-acetate (**7**) taurolipid C (**84**) gallic acid (**87**) cinnamic acid (**97**) ferulic acid (**86**) stigmasterol-*O*-d-glucoside (**107**)	Cytotoxicity In vitro cytotoxicity assay, cell cycle analysis, and apoptosis assay	Cardenolides (kalantubolide A (**38**) and kalantubolide B (**39**)) and bufadienolide glycosides (kalantuboside A (**23**), kalantuboside B (**24**), bryotoxin C (**1**), bersaldegenin-1,3,5-orthoacetate (**6**)**,** bersaldegenin-1-acetate (**7**)) showed strong cytotoxicity against four human tumor cell lines (A549, Cal-27, A2058, and HL-60) with IC_50_ values ranging from 0.01 µM to 10.66 µM. Cardenolides (kalantubolide A (**38**) and kalantubolide B (**39**)) also displayed significant cytotoxicity toward HL-60 tumor cell line. In addition, kalantuboside A (**23**), kalantuboside B (**24**), bryotoxin C (**1**), bersaldegenin-1,3,5-orthoacetate (**6**), and bersaldegenin-1-acetate (**7**) blocked the cell cycle in the G2/M-phase and induced apoptosis in HL-60 cells.	Not reported	[33]
*K. pinnata* Whole plant	bryophyllin A (**1**) bryophyllin B (**2**) bersaldegenin-3-acetate (**8**)	Cytotoxicity In vitro cytotoxicity assay	Bryophyllin A (**1**), bryophyllin B (**2**), and bersaldegenin-3-acetate (**8**) showed potent cytotoxicity effects.	Not reported	[114]
*K. pinnata* Roots	KPB-100 (**81**) KPB-200 (**82**)	Antivirus Virus spread inhibition and virus yield reduction assays of vaccinia virus, and viral cytopathic effect inhibition assay of HHV-2-TK-mutant and VYR assay of HHV-1 wild type	Both compounds are promising targets for synthetic optimization and in vivo study against human alpha herpesvirus 1 and 2 and vaccinia virus. KPB-100 (**122**) strongly inhibited all the tested viruses.	The authors consider that further studies are required to establish the mechanism of action of these compounds.	[69]
*K. pinnata* Flowers	quercetin 3-*O*-α-l-arabinopyranosyl-(1→2)-α-l-rhamnopyranoside (**45**)	Anti-inflammatory Acetic acid-induced abdominal writhing	The flavonoid (1, 3, and 10 mg/kg) produced a dose-related inhibition of the number of acetic acid-induced writhing by 20.5% (44.2 ± 3.1 w), 35.8% (35.7 ± 4.5 w), and 50.5% (27.5 ± 3.5 w), respectively (ID_50_ 9.4 mg/kg), when compared with the vehicle group (55.6 ± 3.3 w). The positive control indomethacin (10 mg/kg) reduced the number of writhings by 56.5% (24.2 ± 3.5 w).	The aglycone quercetin present in the chemical structure of the isolated compound proved to be an anti-inflammatory and immunosuppressive agent. This flavonol has a well-known immunomodulatory effect through the regulation of inflammatory mediators, such as inhibiting cytokine and inducible nitric oxide synthase expression via inhibition of the NF-κβ pathway.	[8]
		Anti-inflammatory Carrageenan-induced pleurisy	The flavonoid (0.3, 1.0, and 3.0 mg/kg) exhibited a dose-related reduction of leukocyte migration by 8.0% (6.9 ± 0.6 leukocytes × 106/mL), 38.8% (4.6 ± 0.2 leukocytes × 106/mL), and 57.2% (3.2 ± 0.3 leukocytes × 106/mL), respectively (ID50 2.0 mg/kg), whereas the treated with dexamethasone (2 mg/kg), positive control group, inhibited by 71.9% (2.1 ± 0.2 leukocytes × 106/mL) when compared with the vehicle-treated group (7.5 ± 0.6 leukocytes × 106/mL).	The reduction in the total leukocyte migration to the pleural cavity induced by carrageenan is dependent on the synthesis/release of the chemoattractant mediators leukotrienes such as LTB4, cytokines IL-1 and TNF-α, and chemokines.	
		Anti-inflammatory Croton oil-induced mice ear edema	Pretreatment with the flavonoid (0.3, 1.0, or 3.0 mg/kg, s.c.) produced a dose-related antiedematogenic effect by 38.2% (=4.2 ± 0.4 mg), 54.4% (=3.1 ± 0.4 mg), and 70.6% (=2.0 ± 0.4 mg), respectively, whereas the treatment with dexamethasone (2 mg/kg) reduced the ear edema by 85.3% (=1.0 ± 0.4 mg) when compared with the vehicle group (=6.8 ± 0.6 mg), with ID50 0.76 mg/kg.	The edema formation is initially mediated by histamine and serotonin and later by the release of prostaglandins. Prostaglandins play an important role in the setting of the cardinal signs of inflammation, pain, heat, redness, edema, and loss of function. The biosynthesis of PGE2, the main inflammatory prostaglandin, involves three key enzymes, phospholipase A2 (PLA2), cyclooxygenase (COX), and PGE synthase (PGES).	
		Anti-inflammatory TNF-α ex vivo measurement	The flavonoid (3.0 mg/kg, s.c.) decreased the TNF-α concentration in pleural exudates by 66.6% (22.6 ± 3.1 pg/mL) when compared to the vehicle group (67.5 ± 4.9 pg/mL), whereas dexamethasone (2 mg/kg, s.c.) reduced the TNF-α concentration by 74.5% (17.2 ± 3.2 pg/mL).	Pretreatment reduced the TNF-α concentration in pleural exudates, suggesting that they produce an anti-inflammatory effect, at least in part, by TNF-α inhibition.	
		Anti-inflammatory In vitro cyclooxygenase (COX) inhibition assay	The flavonoid inhibited both COX-1 and COX-2 in vitro activities (COX-1 IC_50_ = 3.8 × 10^−5^ M (22.1 μg/mL) and COX-2 IC_50_ ≥ 8.4 × 10^−5^ M). The selectivity index was <0.44. The positive control indomethacin also inhibited both COX-1 and COX-2 activities (IC_50_ for COX-1 and COX-2 was 5.9 and 31.2 μg/mL, resp., and SI was 0.19).	Some flavonoids may reduce PGE2 synthesis by inhibiting the activity of these enzymes or by inhibiting the expression of the inflammatory-induced enzymes, COX-2, or microsomal PGES-1.	
*K. pinnata* Leaves	quercetin 3-*O*-α-l-arabinopyranosyl-(1→2)-α-l-rhamnopyranoside (**45**)	Wound healing In vivo rat excision model	A cream containing quercetin 3-*O*-α-l-arabinopyranosyl-(1→2)-α-L rhamnopyranoside (**45**) (0.15%) was developed and topically compared to a cream containing the aqueous extract. Both creams showed a better re-epithelialization and dense collagen fibers compared to control groups.	Wound healing agents can act in the inflammation, cellular proliferation and/or remodeling phases of wound healing. Classic symptoms of inflammation are caused by the release of prostaglandins, leukotrienes and reactive oxygen and nitrogen species. The strong antioxidant activity and in vivo anti-inflammatory activity exhibited by quercetin 3-*O*-α-L-arabinopyranosyl-(1→2)-α-L rhamnopyranoside (45) might explain its healing performance, being considered the main responsible for the wound healing activity of this species.	[67]
	quercetin 3-*O*-α-l-arabinopyranosyl-(1→2)-*O*-α- L-rhamnopyranoside (**45**) kaempferol 3-*O*-α-l-arabinopyranosyl-(1→2)-*O*-α-L-rhamnopyranoside (**56**)	Antioxidant In vitro DPPH and ABTS assays	Kaempferol and quercetin derivatives moderately inhibited XO, while only quercetin derivatives displayed average radical scavenging activity, suggesting that quercetin 3-*O*-α-l-arabinopyranosyl-(1→2)-α-l-rhamnopyranoside (**45**) can be indicated as a specific marker of this species.	Not reported	[71]
		Anti-inflammatory Xanthine oxidase (XO) inhibition assay			
	quercitrin (**43**)	Antianaphylactic Mouse hypersensitization and antigen challenge, OVA-specific IgE measurement, T cell proliferation, cytokine production, mast cell degranulation in the mesentery, and histamine release assay.	Pretreatment with the flavonoid quercitrin (**43**) showed protective effects in death caused by anaphylactic shock. In this study, the treatment conferred resistance to fatal anaphylactic shock in 75% of the animals.	The mechanism by which quercitrin acts its still unknown.	[157]
		Anti-inflammatory Mast cell activation in vitro and allergic airway disease model in vivo	Treatment with quercitrin (**43**) did not affect the tested parameters.		[68]
		Antileishmanial In vitro antiamastigote and antipromastigote acitivity assays	Antiamastigote activity-guided fractionation of ethyl acetate fraction led to the isolation of quercitrin (**43**), which inhibited 93.9% of amastigote growth (100 µg/mL (223 µM). The compound exhibited significant antileishmanial activity.		[130]
	bryophyllin A (**1**) bryophyllin C (**3**)	Insecticidal Third instar larvae of silkworm bioassay	Bryophyllin A (**1**) and bryophyllin C (**3**) showed strong insecticidal activity against third instar larvae of the silkworm (*Bombyx mori*).	The authors suggest that the 1,3,5-orthoacetate moiety played an important role in exhibiting the insecticidal activity.	[112]
	bryophyllin A (**1**) bryophyllin B (**2**) bryophyllin C (**3**)	Antivirus Tumor promoter-induced Epstein-Barr virus (EBV) activation assay	Bryophyllin A (**1**), bersaldegenin 1,3,5-orthoacetate (**6**) and daigremontianin (**4**) showed good inhibitory potential on the Epstein-Barr virus, but bryophyllin A (**1**) was the most effective (IC_50_: 0.4 µM). These results strongly suggest that bufadienolides are potential cancer chemopreventive agents.	Tumor promoters possibly induce EBV activation through the activation of protein kinase C and mitogen-activated protein kinase.	[174]
*K.* × *houghtonni* Leaves	daigremontianin (**4**) bersaldegenin 1,3,5-orthoacetate (**6**)				

## Data Availability

Not applicable.

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
