# Peer review of "A Review of the Popular Uses, Anatomical, Chemical, and Biological Aspects of Kalanchoe (Crassulaceae): A Genus of Plants Known as “Miracle Leaf”"

_molecules, 2023, doi:10.3390/molecules28145574_

Round 1

Reviewer 1 Report

The manuscript under review represents a detailed summary of the literature on Kalanchoe species focusing mainly on their botany, chemistry, pharmacology, and ethnobotany.

The manuscript is well-structured and written. However, there are a few minor issues that should be considered:

On page 12, the first paragraph after the table, the first sentence: “K. pinnata, K. laciniata” should be in italics.

On page 20, “Figure 1”: “Figure 1” should be “Figure 3”. The title of this figure does not correspond to the structures in the figure.

On the same page, “Table 5”: In my opinion, only the plant part should be used in the first column.

On page 23, “Figure 4”: Concerning structures 63-66, the number of each structure should be placed under the respective structure. This rule should also be applied to the other structure figures. Please, consider a re-arrangement of this figure. For example, structures 81 and 82 are lignan glycosides and 101-102 are not “organic/phenolic acids”.

Reviewer 2 Report

In the article entitled: "A review of the popular uses, anatomical and biological aspects of Kalanchoe (Crassulaceae): a genus of plants known as "miracle leaf"". The authors present a critical review of significant findings related to traditional therapeutic uses, botanical characteristics, chemical composition, and pharmacological activity of species of the genus Kalanchoe. Known as the "miracle leaf," these plants have a long history of remarkable healing properties.

The manuscript is interesting, relevant and current.

It would be necessary to write adequately scientific names “K. pinnata y K. laciniata” after table 2.

It would be important for the authors to be able to include in table 3 a column that shows the form of use or preparation of the species

The “Figure 1. Chemical structures of cardiac glycosides from Kalanchoe species". should be figure 4 and should show cardiac glycosides (compounds 1-39).

The “Figure 4. Chemical structures of flavonoids from Kalanchoe species is repeated is the same as” Figure 1. Chemical structures of cardiac glycosides from Kalanchoe species”. Please correct

It would be recommendable that the authors create a table of the biological activity of the species and the isolated compounds that includes the part of the plant used, the type of extract obtained, the model used, the result, and the mechanism of action.
